# State-free Reinforcement Learning

**Mingyu Chen**
Boston University
mingyuc@bu.edu

**Aldo Pacchiano**
Boston University
Broad Institute of MIT and Harvard
pacchian@bu.edu

**Xuezhou Zhang**
Boston University
xuezhouz@bu.edu

## Abstract

In this work, we study the *state-free RL* problem, where the algorithm does not have the states information before interacting with the environment. Specifically, denote the reachable state set by $\mathcal{S}^{\Pi} := \{s| \max_{\pi \in \Pi} q^{P,\pi}(s) > 0\}$, we design an algorithm which requires no information on the state space $S$ while having a regret that is completely independent of $\mathcal{S}$ and only depend on $\mathcal{S}^{\Pi}$. We view this as a concrete first step towards *parameter-free RL*, with the goal of designing RL algorithms that require no hyper-parameter tuning.

## 1 Introduction

Reinforcement learning (RL) studies the problem where an agent interacts with an *unknown environment* to optimize cumulative rewards/losses [Sutton and Barto, 2018]. While the nature of the environment is in principle hidden from the agent, many existing algorithms [Azar et al., 2017, Jin et al., 2018, Zanette and Brunskill, 2019, Zhang et al., 2020, 2021] implicitly require prior knowledge of parameters of the environment, such as the size of the state space, action space, time horizon and so on. Such parameters play a crucial role in these algorithms, as they are used in the construction of variable initializations, exploration bonuses, confidence sets, etc. However, in most real-world problems, these parameters are not known a priori, resulting in the need for the system designer to perform hyper-parameter tuning in a black-box fashion, which is known to be extremely costly in RL compared to their supervised learning counterparts [Pacchiano et al., 2020]. In supervised learning algorithms, selecting among $M$ hyper-parameters only degrades the sample complexity by a factor of $O(\log(M))$. In contrast, in RL problems it will incur a $O(\sqrt{M})$ multiplier on the regret, making hyper-parameter tuning prohibitively expensive. This is one of the major roadblocks to broader applicability of RL to real-world scenarios.

Motivated by the above observation, we propose and advocate for the study of **parameter-free reinforcement learning**, i.e. the design of RL algorithms that have no or as few hyper-parameters as possible, with the eventual goal of eliminating the need for heavy hyper-parameter tuning in practice. As a concrete first step, in this paper, we focus on the problem of *state-free RL* in tabular MDPs. In particular, we will show that there exist state-free RL algorithms which do not require the state space $S$ as an input parameter to the algorithm, nor do their regret scale with the innate state space size $|S|$. In particular, we design a black-box reduction framework called State-Free Reinforcement Learning (SFRL). Given any existing RL algorithm for stochastic or adversarial MDPs, this framework can transform it into a state-free RL algorithm through a black-box reduction. We also show that the same framework can be adapted to induce action-free and horizon-free algorithms, the three of which now makes a tabular MDP algorithm completely parameter-free, i.e. it requires no input parameters whatsoever, and their regret bound automatically adapt to the intrinsic complexity of the problem.

The rest of the paper is organized as follows. Following the discussion of related works and problem formulation, we start by discussing the technical challenges of state-free learning and why existing algorithmic and analysis framework are not able to achieve it (Section 4). Built upon these insights, we propose an intuitive black-box reduction framework `SF-RL`, that transforms any RL algorithm

38th Conference on Neural Information Processing Systems (NeurIPS 2024).

into a state-free RL algorithm, albeit incurring a multiplicative cost to the regret (Section 5). Further improvements are then made to eliminate the additional cost through a novel confidence interval design, which can be of independent interest (Section 6).

## 2 Related Works

**Parameter-free algorithms:** Acknowledgedly, parameter-free learning is not a new concept and has been studied extensively in the optimization and online learning community. Parameter-free algorithms refer to algorithms that do not require the learner to specify certain hyperparameters in advance. These algorithms are appealing in both theory and practice, considering that tuning algorithmic parameters is a challenging task [Bottou, 2012, Schaul et al., 2013]. The types of hyperparameters to "set free" varies depending on the specific problem. For example, for online learning and bandit problems, the hyperparameters are considered as the scale bound of the losses [De Rooij et al., 2014, Orabona and Pál, 2018, Duchi et al., 2011, Chen and Zhang, 2023], or the range of the decision set [Orabona and Pál, 2016, Cutkosky and Orabona, 2018, Zhang et al., 2022, van der Hoeven et al., 2020]; for neural network optimization, the hyperparameters can be the learning rate of the optimizer [Defazio and Mishchenko, 2023, Carmon and Hinder, 2022, Ivgi et al., 2023, Cutkosky et al., 2024, Khaled and Jin, 2024]; for model selection, the hyperparameters are the choice of the hypothesis class [Foster et al., 2017, 2019].

Surprisingly, the reinforcement learning (RL) community has overlooked the concept of parameter-free learning almost entirely. To the best of our knowledge, the only related work is from Chen and Zhang [2024], where the authors proposed an algorithm that adapts to the scale of the losses in the setting of adversarial MDPs. In this work we focus on the problem of developing parameter-free RL algorithms where the parameter to be focused on are those related to the environment transition, particularly the state space. Almost all RL algorithms assume knowledge of the state-space. For example, existing UCB-based reinforcement learning algorithms [Azar et al., 2017, Jin et al., 2018, Zanette and Brunskill, 2019, Zhang et al., 2020, 2021] make use of the state space size to construct the UCB bonus. When the state space is unknown, it is unclear whether these algorithms can still build a valid UCB bonus that ensures optimism and achieve bounded regrets.

**Instance-dependent algorithms:** *Instance-dependent learning* is a closely related concept to parameter-free learning. Instance-dependent algorithms dynamically adjust to the input data they find, and achieve a regret that not only scaling with the number of iterations $T$, but also adapt to certain "measures of hardness" of the environment. Such algorithms perform better than the worst-case regret if the environment is "benign". In reinforcement learning, the most common "measures of hardness" considered in the community are *Variance* [Zanette and Brunskill, 2019, Zhou et al., 2023, Zhang et al., 2023, Zhao et al., 2023] and *Gap* [Simchowitz and Jamieson, 2019, Xu et al., 2021, Dann et al., 2021, Jonsson et al., 2020, Wagenmaker et al., 2021, Tirinzoni et al., 2021], both related to the reward of the environment. Specifically, variance-dependent algorithms provide regret bounds that scale with the underlying conditional variance of the $Q^\star$ function. Gap-dependent algorithms provide regret bounds of order $\tilde{\mathcal{O}}(\log T / \text{gap}(s, a))$ where the gap notion is defined as the difference of the optimal value function and the $Q^\star$-function at a sub-optimal action $V^\star(s) - Q^\star(s, a)$ [Dann et al., 2021]. Additionally, some studies consider problems similar to ours, that is, how to adapt to the "measure of hardness" of the state space. Given an initial state space, Fruit et al. [2018] proposes an algorithm that adapts to the size of the reachable state space, resulting in improved performance when the initial state space is vacuous.

The difference between instance-dependent algorithm and parameter-free algorithm is subtle. Both family of algorithms have the capability to adapt to the input data, allowing them to sequentially tune the hyperparameters and ultimately converge to the optimal hyperparameters inherent in the data. Consequently, when the number of iterations becomes sufficiently large, both instance-dependent algorithms and parameter-free algorithms tend to provide the same theoretical guarantees. However, this does not mean that the two types of algorithms are the same. The most significant difference is that instance-dependent algorithms require appropriate **hyper-parameters initialization**. Taking state-space adaptability as an example. Let $N$ represent the true number of states. An instance-dependent algorithm must be provided with an initial value $M \geq N$. If this value is invalid, i.e., $M < N$, the algorithm will fail to function properly. Moreover, the regret of instance-dependent algorithms is typically related to the initial input, even though this dependency may fade away

as the number of iterations increases. This is also why we cannot simply set $M$ to infinity in an instance-dependent algorithm and call it parameter-free, that is, the regret of an instance-dependent algorithm always includes some burn-in terms that scale with $M$. As $M$ goes to infinity, the burn-in term eventually dominates. In this sense, parameter-free learning is a strictly harder problem than instance-dependent learning.

# 3  Problem Formulation

**Markov Decision Process:** This paper focuses on the episodic MDP setting with finite horizon, unknown transition, and bandit feedback. A MDP is defined by a tuple $\mathcal{M} = (\mathcal{S}, \mathcal{A}, H, P)$, where $\mathcal{S} = \{1, \ldots, S\}$ denotes the state space, $\mathcal{A} = \{1, \ldots, A\}$ denotes the action space, and $H$ denotes the planning horizon. $P : S \times A \times S \to [0, 1]$ is an unknown transition function where $P(s'|s, a)$ is the probability of reaching state $s'$ after taking action $a$ in state $s$. For every $t \in [T]$, we define $\ell_t : \mathcal{S} \times \mathcal{A} \to [0, 1]$ as the loss function. In stochastic MDPs, the loss function $\ell_t$ is drawn from a time-independent distribution. In adversarial MDPs, the loss function $\ell_t$ is determined by the adversary, which can depend on the player's actions before $t$. The learning proceeds in $T$ episodes. In each episode $t$, the learner starts from state $s_1$ and decides a stochastic policy $\pi_t \in \Pi : \mathcal{S} \times \mathcal{A} \to [0, 1]$ with $\pi_t(a|s)$ being the probability of taking action $a$ in state $s$. Afterwards, the learner executes the policy in the MDP for $H$ steps and observes a state-action-loss trajectory $(s_1, a_1, \ell_t(s_1, a_1), \ldots, s_H, a_H, \ell_t(s_H, a_H))$ before reaching the end state $s_{H+1}$. With a slight abuse of notation, we assume $\ell_t(\pi) = \mathbb{E}[\sum_{h \in [H]} \ell_t(s_h, a_h)|P, \pi]$. The performance is measured by the regret, which is defined by

$$\mathbb{R}(T) = \sum_{t=1}^{T} \ell_t(\pi_t) - \min_{\pi \in \Pi} \sum_{t=1}^{T} \ell_t(\pi).$$

Without loss of generality, we consider a layered-structure MDP: the state space is partitioned into $H + 2$ horizons $S_0, \ldots, S_{H+1}$ such that $S = \cup_{h=1}^{H} S_h$, $\emptyset = S_i \cap S_j$ for every $i \neq j$, $S_0 = \{s_0\}$ and $S_{H+1} = \{s_{H+1}\}$.

**Occupancy measure:** Given the transition function $P$ and a policy $\pi$, the occupancy measure $q : \mathcal{S} \times \mathcal{A} \to [0, 1]$ induced by $P$ and $\pi$ is defined as

$$q^{P,\pi}(s, a) = \sum_{h=1}^{H} \mathbb{P}\left(s_h = s, a_h = a | P, \pi\right).$$

Using occupancy measures, the MDP problem can be interpreted in a way that makes it similar to Multi-armed Bandit (MAB) because for any policy $\pi$, the loss can be expressed as

$$\ell_t(\pi) = \sum_{s \in [S]} \sum_{a \in [A]} q^{P,\pi}(s, a) \ell_t(s, a) = \langle q^{P,\pi}, \ell_t \rangle.$$

Using this formula the regret can be written as $\mathbb{R}(T) = \sum_{t=1}^{T} \langle q^{P,\pi_t} - q^{P,\pi_\star}, \ell_t \rangle$.

**State-free RL:** We say a state $s \in \mathcal{S}$ is *reachable* to a policy set $\Pi$ if there exists a policy $\pi \in \Pi$ such that $q^{P,\pi}(s) > 0$. We further define $\mathcal{S}^\Pi = \{s \in \mathcal{S} | \max_{\pi \in \Pi} q^{P,\pi}(s) > 0\}$ to represent all the reachable states to $\Pi$ in $\mathcal{S}$. The formal definition *state-free* algorithm is proposed below.

**Definition 3.1.** *(State-free algorithm): We say a RL algorithm is state-free if given any policy set $\Pi$, the regret bound for the algorithm can be adaptive to $|\mathcal{S}^\Pi|$ and independent to $|\mathcal{S}|$, without any knowledge of the state space a priori.*

At first glance, designing state-free algorithms appears straightforward: if the learner had access to the transition $P$, it can compute $\max_{\pi \in \Pi} q^{P,\pi}(s)$ for every state $s \in \mathcal{S}$ and then remove all the unreachable states, thereby reducing the state space $\mathcal{S}$ to $\mathcal{S}^\Pi$. Through this reduction, any existing MDP algorithm can be made state-free. However, such a method is infeasible since $P$ is always unknown in practice. Without the knowledge of $P$, it becomes challenging or even impossible to determine whether a state is reachable or not. In the following section, we elaborate on the technical challenges of the problem for both stochastic and adversarial loss settings.

# 4 Technical challenges

In this section, we explain the technical challenges for the state-free learning. Specifically, we consider a weakened setup. We assume for a moment that the algorithm has access to the state space $\mathcal{S}$ but not the reachable space $\mathcal{S}^\Pi$. It is clear that this setup is weaker than the state-free definition, as in the state-free setting, the information about $\mathcal{S}$ is also unknown.

We start with the stochastic setting, where the loss function $\ell_t$ is sampled by a time-independent distribution for all $t \in [T]$. As the most prominent setting in RL research, numerous works have since been devoted to improving the regret guarantee and the analysis framework [Brafman and Tennenholtz, 2003, Kakade, 2003, Jaksch et al., 2010, Azar et al., 2017, Jin et al., 2018, Dann et al., 2017, Zanette and Brunskill, 2019, Bai et al., 2019, Zhang et al., 2020, 2021, Ménard et al., 2021, Li et al., 2021, Domingues et al., 2021]. Surprisingly, although existing works have not mentioned the state-free concept explicitly, we find that some algorithms can almost achieve state-free learning without algorithmic modifications. In particular, we have

**Proposition 4.1.** *For stochastic MDPs, UCBVI [Azar et al., 2017] is a weakly state-free algorithm, that is, with only the knowledge of $\mathcal{S}$, the regret guarantee of UCBVI is adaptive to $|\mathcal{S}^\Pi|$ and independent to $|\mathcal{S}|$, except in the logarithmic terms.*

Proposition 4.1 offers some positive insights into existing algorithms. The source of the log-dependence on $|\mathcal{S}|$ is straight-forward: the analysis of RL algorithms needs to ensure that concentration inequalities hold for all states with probability at least $1 - \delta$. At this point, since the events among the states are independent from each other, the algorithms have to take a union bound across the state space to make concentration holds simultaneously in all states, which implies that the confidence level $\delta$ needs to be divided by $|\mathcal{S}|$. This leads to a regret guarantee that scale with $\log(|\mathcal{S}|)$.

**Remark 4.2.** *In Appendix A, we propose a simple technique to get rid of the log-dependence on $|\mathcal{S}|$ under the UCBVI framework. The key is to allocate the confidence for each visited $(s, a)$ pairs sequentially, instead of applying a uniform confidence allocation across all states in $|\mathcal{S}|$. We further show that such a method removes the need of $\mathcal{S}$ information in the algorithm design. Based on this method, it is suffices to conclude that (a modified version of) UCBVI is a state-free algorithm.*

We now turn our attention to the adversarial setting. In adversarial MDPs, the loss is determined by the adversary and can be depend on previous actions. Adversarial MDPs have been studied extensively in recent years Jin et al. [2019], Dai et al. [2022], Lee et al. [2020], Luo et al. [2021]. Given the positive results for stochastic MDPs, one might hope that existing adversarial MDP algorithms can naturally achieve a state-free regret guarantees. Unfortunately, this is not the case. In particular, we have the following observation.

**Observation 4.3.** *(Informal) In adversarial MDPs, using the existing algorithms and analysis framework, the regret guarantee cannot escape a polynomial-level dependence on $|\mathcal{S}|$.*

Here we briefly explain Observation 4.3. In all prior works on adversarial MDPs, the analysis relies on bounding the gap between the approximation transition function $\hat{P}$ and the true one $P$, i.e., $\sum_{s \in \mathcal{S}^\Pi \times \mathcal{A}} \|\hat{P}(\cdot|s, a) - P(\cdot|s, a)\|_1$. In this case, for any state $s' \in \mathcal{S}$, regardless of whether $s'$ is reachable or not, the estimation error $|\hat{P}(s'|s, a) - P(s'|s, a)|$ may remain non-zero for all $(s, a)$ pairs. Consequently, the larger the $|\mathcal{S}|$, the greater the inaccuracy of the transition estimation. At this point, one may wonder if the learner can directly set $\hat{P}(s'|s, a) = 0$ for all unvisited $s' \in \mathcal{S}$, so that $|\hat{P}(s'|s, a) - P(s|s, a)|$ is always zero when $s'$ is unreachable. However, since the learner does not have the knowledge of $P$, it is impossible to determine whether a state is unreachable, even if the learner has never visited the state before. In this regard, if $s'$ is actually reachable, the transition estimator will become invalid. Such dilemma constitutes the main challenge of the problem.

The above offers some high-level intuitions into the complexity of designing state-free algorithms. In the next section, we introduce our new algorithms for state-free RL that operate without prior knowledge of $\mathcal{S}$.

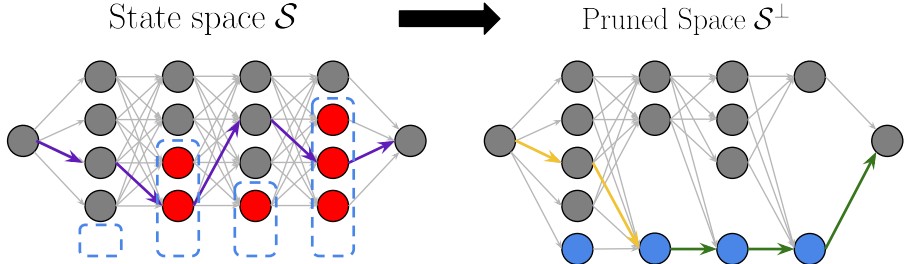

Figure 1: An illustration of the mapping between the state space $\mathcal{S}$ and the pruned space $\mathcal{S}^\perp$. The left side represents the original state space $\mathcal{S}$, where grey nodes denote the states in $\mathcal{S}^\perp$ and red nodes denote the others. The right side is the corresponding pruned space $\mathcal{S}^\perp$, where blue nodes denote the auxiliary states $\{s_h^\perp\}_{h \in [H]}$. Given the structure, for any trajectory in space $\mathcal{S}$ (purple arrows), we can find a dual trajectory (yellow and green arrows) in the pruned space.

## 5  Black-box reduction for State-free RL

In this section, we outline the main contribution of the paper. To generalize our results further, we denote the $\epsilon$-reachable state space as $\mathcal{S}^{\Pi,\epsilon} = \{s \in \mathcal{S} \mid \max_{\pi \in \Pi} q^{P,\pi}(s) > \epsilon\}$. By definition $\mathcal{S}^\Pi = \mathcal{S}^{\Pi,0}$. Our algorithm SF-RL is illustrated in Algorithm 1. The algorithm maintains a pruned state space, denoted by $\mathcal{S}^\perp$, which includes all the identified $\epsilon$-reachable states and $H$ additional auxiliary states. Throughout $t = 1, \ldots, T$, SF-RL first obtains the policy $\pi_t^\perp \in \Pi^\perp : \mathcal{S}^\perp \to \Delta(\mathcal{A})$ from a black-box adversarial MDP algorithm, namely ALG, which operates on $\mathcal{S}^\perp$. Then, by playing an arbitrary action on states not in $\mathcal{S}^\perp$ compatible with $\Pi$, it extends the pruned policy $\pi_t^\perp$ to $\pi_t \in \Pi$, which is defined over the domain $\mathcal{S}$, and then receives the trajectory $o_t$ after playing $\pi_t$. Given the trajectory, if there exists a new state $s \notin \mathcal{S}^\perp$ that can be confirmed to be $\epsilon$-reachable, the algorithm will update $\mathcal{S}^\perp$ and restart the ALG subroutine. Otherwise, SF-RL pretends that the trajectory was produced only by interacting with $\mathcal{S}^\perp$, and sends the pruned trajectory $o_t^\perp$ back to ALG.

The key novelty of the algorithm lies in the design of the pruned space $\mathcal{S}^\perp$ and trajectory $o_t^\perp$. For every $h \in [H]$, the auxiliary state $s_h^\perp$ represents the collection of states $\mathcal{S}_h \setminus \mathcal{S}_h^\perp$. These behave as "absorbing" states, coalescing all the transitions to states not in $\mathcal{S}^\perp$. Given the pruned space $\mathcal{S}^\perp$ and $o_t$, we can build the pruned trajectory $o_t^\perp = \{s_h', a_h', \ell_t'(s_h', a_h')\}_{h \in [H]}$. Specifically, $o_t^\perp$ can be split into two parts based on the horizon that first encounters the state not in $\mathcal{S}^\perp$, i.e., $h = \arg\max_h \{s_{1:h} \in \mathcal{S}^\perp\}$. For the state-action-loss pairs before the split horizon, we set $o_t^\perp$ to be the same as in $o_t$. Otherwise, we let the states to be the corresponding auxiliary states, the actions to be $\pi^\perp$, which represents an arbitrary action, and the loss to be zero. An illustration of the design is provided in Figure 1.

In order to analyze the performance of our black-box SF-RL algorithm we assume the input ALG comes equipped with a regret bound,

**Assumption 5.1.** *(Regret guarantee for black-box algorithm ALG): With probability $1 - \delta$, for all $K > 0$, the regret guarantee for ALG following $K$ epochs of interaction with MDP $\mathcal{M} = (\mathcal{S}, \mathcal{A}, H, P)$ is bounded by*

$$\mathbb{R}^{ALG}(K) \le reg\left(|\mathcal{S}|, |\mathcal{A}|, H, \log\left(H|\mathcal{S}||\mathcal{A}|K/\delta\right)\right)\sqrt{K},$$

*where $reg(|\mathcal{S}|, |\mathcal{A}|, H, \delta)$ is a coefficient that depends polynomially on $|\mathcal{S}|, |\mathcal{A}|, H$ and $\log(H|\mathcal{S}||\mathcal{A}|K/\delta)$. Moreover, we assume that the coefficient $reg$ is non-decreasing as $|\mathcal{S}|, |\mathcal{A}|, H, K, 1/\delta$ increase.*

This definition works for most algorithms in both stochastic and adversarial environments, e.g., for stochastic MDPs, by setting ALG as UCBVI Azar et al. [2017], the coefficient can be set as $\mathcal{O}(H\sqrt{|\mathcal{S}||\mathcal{A}|\log(|\mathcal{S}||\mathcal{A}|K/\delta)})$; for adversarial MDPs, by setting ALG as UOB-REPS Jin et al. [2019], the coefficient can be set as $\mathcal{O}(H|\mathcal{S}|\sqrt{|\mathcal{A}|\log(|\mathcal{S}||\mathcal{A}|K/\delta)})$. If $reg(\cdot)$ is the regret coefficient function for input algorithm ALG, the regret bound for Algorithm 1 satisfies

**Theorem 5.2.** *With probability* $1 - \delta$*, the state-free algorithm* SF-RL *achieves regret bound* [1]

$$\mathbb{R}(T) \le \mathcal{O}\left(reg\left(|\mathcal{S}^{\Pi,\epsilon}| + H, |\mathcal{A}|, H, \log\left(H|\mathcal{S}^{\Pi,\epsilon}||\mathcal{A}|T/\delta\right)\right)\sqrt{|\mathcal{S}^{\Pi,\epsilon}|T} + \epsilon H|\mathcal{S}^{\Pi}|T\right).$$

As shown in Theorem 5.2, the regret bound consists of two terms. The first term is $\sqrt{|\mathcal{S}^{\Pi,\epsilon}|}$ times the regret of the black-box algorithm ALG over $T$ iterations, while the second term can be considered as the regret incurred by the barely reachable states we have disregarded. The trade-off between these two terms is reasonable because it is impossible to discard states that are not $\epsilon$-reachable without incurring any cost. By setting $\epsilon = 0$, Theorem 5.2 immediately provides a regret bound adaptive to the unknown state size $|\mathcal{S}^{\Pi}|$ [2]. Additionally, we remark that SF-RL does not require any prior knowledge about the state space in the algorithm design, which means that SF-RL is state-free by design. Below we discuss the main steps in establishing the above result.

**Proof Highlight:** We first define $P^{\perp} : \mathcal{S}^{\perp} \times \mathcal{A} \times \mathcal{S}^{\perp} \to [0, 1]$ be the underlying transition function on the pruned space $\mathcal{S}^{\perp}$. Specifically, for every $h \in [H]$, we set

$$P^{\perp}(s'|s,a) = P(s'|s,a), \qquad\qquad \forall (s,a,s') \in \mathcal{S}_h^{\perp} \setminus \{s_h^{\perp}\} \times \mathcal{A} \times \mathcal{S}_{h+1}^{\perp} \setminus \{s_{h+1}^{\perp}\}$$

$$P^{\perp}(s'|s,a) = 1 - \sum_{s^{\dagger} \in \mathcal{S}_{h+1}^{\perp} \setminus \{s_{h+1}^{\perp}\}} P(s^{\dagger}|s,a), \qquad \forall (s,a,s') \in \mathcal{S}_h^{\perp} \setminus \{s_h^{\perp}\} \times \mathcal{A} \times \{s_{h+1}^{\perp}\}$$

$$P^{\perp}(s'|s,a) = \mathbb{1}\{s' = s_{h+1}^{\perp}\}, \qquad\qquad \forall (s,a,s') \in s_h^{\perp} \times \mathcal{A} \times \mathcal{S}_{h+1}^{\perp}.$$

Similarly, we define $\ell_t^{\perp} : \mathcal{S}^{\perp} \times \mathcal{A} \to [0, 1]$ to be the loss function on the pruned space $\mathcal{S}^{\perp}$, which satisfies that

$$\ell_t^{\perp}(s,a) = \begin{cases} \ell_t(s,a), & s \notin \{s_h^{\perp}\}_{h\in[H]} \\ 0, & \text{otherwise} \end{cases}, \ \forall (s,a) \in \mathcal{S}^{\perp} \times \mathcal{A}$$

Note that the tuple $\mathcal{M}^{\perp} = (\mathcal{S}^{\perp}, \mathcal{A}, H, P^{\perp})$ is a well-defined MDP. In what follows we use the subscript $t$ to represent the estimators of the objects above at the beginning of epoch $t$, e.g., $\mathcal{S}_t^{\perp}, P_t^{\perp}$. The key lemma of the proof is the following.

**Lemma 5.3.** *It suffices to consider* $o_t^{\perp}$*, which is the pruned trajectory corresponding to* $o_t$*, as an instance by executing policy* $\pi_t^{\perp}$ *on the pruned space* $\mathcal{S}^{\perp}$ *with transition function* $P_t^{\perp}$ *and loss* $\ell_t^{\perp}$*.*

Lemma 5.3 reveals how the black-box algorithm ALG can work. By Assumption 5.1, in order to make the regret independent to $|\mathcal{S}|$, we let ALG perform on the pruned MDP $\mathcal{M}^{\perp}$ instead of $\mathcal{M}$. However, since $\mathcal{M}^{\perp}$ is actually a "virtual" MDP, we cannot account for ALG's interaction with it. To ensure that ALG can be updated correctly, in Lemma 5.3 we show the pruned trajectory $o_t^{\perp}$ can be viewed as a trajectory from executing policy $\pi_t^{\perp}$ on $\mathcal{M}^{\perp}$ and $\ell_t^{\perp}$. Denote the optimal in-hindsight policy as $\pi_\star = \arg\min_{\pi \in \Pi} \sum_{t=1}^{T} \langle \ell_t, \pi \rangle$ and let $\pi_\star^{\perp}$ be the corresponding policy on the pruned space, we start by the regret decomposition below.

$$\mathbb{R}(T) = \underbrace{\sum_{t=1}^{T} \langle q^{P_t^{\perp},\pi_t^{\perp}} - q^{P_t^{\perp},\pi_\star^{\perp}}, \ell_t^{\perp} \rangle}_{\text{\textcircled{1}}} + \underbrace{\sum_{t=1}^{T} \langle q^{P,\pi_t} - q^{P,\pi_\star}, \ell_t \rangle - \sum_{t=1}^{T} \langle q^{P_t^{\perp},\pi_t^{\perp}} - q^{P_t^{\perp},\pi_\star^{\perp}}, \ell_t^{\perp} \rangle}_{\text{\textcircled{2}}}.$$

Here, term \textcircled{1} represents ALG's regret and term \textcircled{2} corresponds to the sum of the error incurred by the difference between $\mathcal{S}$ and $\mathcal{S}^{\perp}$.

**Bounding** \textcircled{1}**:** Let intervals $\mathcal{I}_1, \ldots, \mathcal{I}_M$ be a partition of $[T]$, such that $P_t^{\perp} = P_{(m)}^{\perp}$ for all $t \in \mathcal{I}_t$. We can rewrite the regret as

$$\text{\textcircled{1}} = \sum_{m=1}^{M} \sum_{t\in\mathcal{I}_m} \langle q^{P_{(m)}^{\perp},\pi_t^{\perp}} - q^{P_{(m)}^{\perp},\pi_\star^{\perp}}, \ell_t^{\perp} \rangle.$$

---

[1] For brevity we consider $|\mathcal{S}^{\pi}| \ll T$. Detailed regret is provided in the appendix.

[2] Note that the optimal choice of $\epsilon$ should not be 0 in general, e.g., by setting ALG as UCBVI and $\epsilon = 1/T$, SF-RL achieves regret $\mathcal{O}(H|\mathcal{S}^{\Pi,1/t}|\sqrt{|\mathcal{A}|T} + H|\mathcal{S}^{\Pi}|)$. Such regret will be much smaller than the regret under $\epsilon = 0$ when $|\mathcal{S}^{\Pi,1/t}| \le |\mathcal{S}^{\Pi}|$.

Since $\sum_{m=1}^{\infty} \delta/2m^2 \leq \delta$, using Lemma 5.3 and Assumption 5.1, $(1)$ can be bounded below with probability at least $1 - \delta$.

$$(1) \leq \sum_{m=1}^{M} reg\left(|\mathcal{S}_{(m)}^{\perp}|, |\mathcal{A}|, H, \log\left(2m^2 H |\mathcal{S}_{(m)}^{\perp}||\mathcal{A}||\mathcal{I}_m|/\delta\right)\right) \sqrt{|\mathcal{I}_m|}.$$

Now we continue the proof by bounding $M$ and $\mathcal{S}_{(m)}^{\perp}$. As in SF-RL, a state $s \in \mathcal{S}$ will be added in the pruned space if it satisfies $\sum_{j=1}^{t} \mathbb{1}_j\{s\}/2 - \log(2H^2 t^2/\delta) - 1/2 > \epsilon t$. Such a design ensures that all states added in $S^{\perp}$ are at least $\epsilon$-reachable, which is formalized in the following lemma.

**Lemma 5.4.** *With probability $1 - \delta$, for every state $s \in \mathcal{S}$, it will be added in $\mathcal{S}^{\perp}$ only if the state is $\epsilon$-reachable, i.e., $\max_{\pi \in \Pi} q^{P,\pi}(s) > \epsilon$.*

By Lemma 5.4, it suffices to say that $|\mathcal{S}_{(m)}^{\perp}| \leq |\mathcal{S}^{\pi,\epsilon}| + H$ for all $m \in [M]$ and $M \leq |\mathcal{S}^{\pi,\epsilon}|$, as the states not in $\mathcal{S}^{\pi,\epsilon}$ cannot be added in $\mathcal{S}^{\perp}$. This result also implies that ALG can be restarted at most $|\mathcal{S}^{\Pi,\epsilon}|$ times, thus $M \leq |\mathcal{S}^{\Pi,\epsilon}| + 1$. Given the above, we can finally bound $(1)$ by

$$(1) \leq reg\left(|\mathcal{S}^{\Pi,\epsilon}| + H, |\mathcal{A}|, H, \log\left(2H(|\mathcal{S}^{\Pi,\epsilon}| + H)^3 |\mathcal{A}|T/\delta\right)\right) \sum_{m=1}^{M} \sqrt{|\mathcal{I}_m|}$$

$$\leq reg\left(|\mathcal{S}^{\Pi,\epsilon}| + H, |\mathcal{A}|, H, \log\left(2H(|\mathcal{S}^{\Pi,\epsilon}| + H)^3 |\mathcal{A}|T/\delta\right)\right) \sqrt{|\mathcal{S}^{\Pi,\epsilon}|T}.$$

**Bounding $(2)$:** The proof relies on the following lemma.

**Lemma 5.5.** *Given the pruned space $\mathcal{S}^{\perp}$ and the corresponding transition $P^{\perp}$, for any policy $\pi$, there is*

$$0 \leq \langle q^{P,\pi}, \ell_t \rangle - \langle q^{P^{\perp},\pi^{\perp}}, \ell_t^{\perp} \rangle \leq H \sum_{s \in \mathcal{S}^{\Pi}} q^{P,\pi}(s) \mathbb{1}\{s \notin \mathcal{S}^{\perp}\}.$$

Using Lemma 5.5, we immediately have

$$(2) \leq \left(\sum_{t=1}^{T} \langle q^{P,\pi_t}, \ell_t \rangle - \sum_{t=1}^{T} \langle q^{P_t^{\perp},\pi_t^{\perp}}, \ell_t^{\perp} \rangle\right) \leq H \sum_{t=1}^{T} \sum_{s \in \mathcal{S}} q^{P,\pi_t}(s) \mathbb{1}_t\{s \notin \mathcal{S}_t^{\perp}\}.$$

It then suffices to bound the right hand side of the inequality. Denote by $X_t = \sum_{s \in \mathcal{S}} \mathbb{1}_t\{s\} \mathbb{1}_t\{s \notin \mathcal{S}_t^{\perp}\}$. By definition, we have $X_t \in [0, H]$ and $\mathbb{E}[X_t|\mathcal{F}_{t-1}] = \sum_{s \in \mathcal{S}} q^{P,\pi}(s) \mathbb{1}_t\{s \notin \mathcal{S}_t^{\perp}\}$. Using Lemma B.1 in the appendix, with probability $1 - \delta$, we have

$$(2) \leq 2H \sum_{s \in \mathcal{S}} \sum_{t=1}^{T} \mathbb{1}_t\{s\} \mathbb{1}_t\{s \notin \mathcal{S}_t^{\perp}\} + 2H^2 \log\left(\frac{1}{\delta}\right).$$

As in SF-RL, if a state has been visited $2\epsilon t + 2\log(2H^2 T^2/\delta) + 2$ times, the state will be added in $\mathcal{S}^{\perp}$, which means $\mathbb{1}_j\{s\} \mathbb{1}_j\{s \notin \mathcal{S}_j^{\perp}\}$ will be 0 for the rest $j > t$. This implies that $\sum_{t=1}^{T} \mathbb{1}_t\{s\} \mathbb{1}_t\{s \notin \mathcal{S}_t^{\perp}\}$ is at most $2\epsilon T + 2\log(2H^2 t^2/\delta) + 2$ for all $s \in \mathcal{S}$. Moreover, if a state is not reachable by any policy in $\Pi$, we always have $\sum_{t=1}^{T} q^{P,\pi}(s) \mathbb{1}_t\{s \notin \mathcal{S}_t^{\perp}\} = 0$. Therefore, we can conclude that $(2) \leq 2\epsilon H |\mathcal{S}^{\Pi}|T + 2H^2 |\mathcal{S}^{\Pi}| \log(2H^2 T^2/\delta) + 2|\mathcal{S}^{\Pi}| + 2H^2 \log(1/\delta)$. Finally, realizing that we have conditioned on the events stated in Assumption 5.1, Lemma 5.4 and Lemma C.1, which happens with probability at least $1 - 3\delta$. By combining $(1)$ and $(2)$ and rescaling $\delta$, we complete the proof.

**Remark 5.6.** *Interestingly, the SF-RL framework can be extended to build horizon-free and action-free algorithm, that is, algorithms that do not require the horizon length (when the horizon length is variable) and action space as input parameters. Specifically, given $\mathcal{S}^{\perp}$, we denote $H^{\perp}$ by the maximum horizon corresponding to the states in $\mathcal{S}^{\perp} \setminus \{s_h^{\perp}\}_{h=1}^{H}$, which represents the maximum horizon index among identified $\epsilon$-reachable states. We further denote $\mathcal{A}^{\perp}$ by the actions corresponding to $\mathcal{S}^{\perp} \setminus \{s_h^{\perp}\}_{h=1}^{H}$. When $\mathcal{S}^{\perp}$ is updated, we let ALG restart with hyper-parameters $(\mathcal{S}_{1:H^{\perp}}^{\perp}, \mathcal{A}^{\perp}, H^{\perp}, P_{1:H^{\perp}}^{\perp})$, where $\mathcal{S}_{1:H^{\perp}}^{\perp}$ and $P_{1:H^{\perp}}^{\perp}$ represent the states and transitions within the first $H^{\perp}$ horizons of $\mathcal{S}^{\perp}$ and $\mathcal{P}^{\perp}$. By using the sub-trajectory of $o_t^{\perp}$ within the first $H^{\perp}$ horizons as the trajectory input of ALG, it suffices to note that Lemma 5.3 still holds, thereby the black-box reduction also works. With such extension, SF-RL requires no hyper-parameter from the environment and can be regarded as completely parameter-free.*

---

**Algorithm 1** Black-box Reduction for State-free RL (`SF-RL`)

---

1: **Input:** action space $\mathcal{A}$, horizon $H$, black-box algorithm `ALG`, confidence $\delta$, pessimism level $\epsilon$
2: **for** $t = 1$ **to** $T$ **do**
3:     Receive policy $\pi_t^\perp : \mathcal{S}^\perp \to \Delta(\mathcal{A})$ from `ALG`
4:     Derive $\pi_t : \mathcal{S} \to \Delta(\mathcal{A})$ such that $\pi_t(\cdot|s) = \begin{cases} \pi_t^\perp(\cdot|s), & s \in \mathcal{S}^\perp \\ \pi^\perp(\cdot|s), & \text{otherwise} \end{cases}$
5:     Play policy $\pi_t$, receive trajectory $o_t = \{s_h, a_h, \ell_t(s_h, a_h)\}_{h \in [H]}$
6:     **if** $\exists s \in o_t, s.t., s \notin \mathcal{S}^\perp, \sum_{j=1}^t \mathbb{1}_j \{s\}/2 - \log\left(2H^2 t^2/\delta\right)/2 - 1/2 > \epsilon t$ **then**
7:         Update $\mathcal{S}^\perp = \mathcal{S}^\perp \cup \{s \in \mathcal{S} : \sum_{j=1}^t \mathbb{1}_j \{s\}/2 - \log\left(2H^2 t^2/\delta\right)/2 - 1 > \epsilon t\}$
8:         Update policy set $\Pi^\perp = \begin{cases} \pi^\perp(\cdot|s) \in \Pi(\cdot|s), & s \in S^\perp \setminus \{s_h^\perp\}_{h \in [H]} \\ \pi^\perp(\cdot|s) \in \{a^\perp\}, & \text{otherwise} \end{cases}$
9:         Restart `ALG` with state space $\mathcal{S}^\perp$, action space $\mathcal{A}$, policy set $\Pi^\perp$ and confidence $\frac{\delta}{2|\mathcal{S}^\perp|^2}$
10:    **else**
11:        Derive the pruned trajectory $o_t^\perp = \{s_h', a_h', \ell_t'(s_h', a_h')\}_{h \in [H]}$ such that

$$s_h' = \begin{cases} s_h, & s_{1:h} \in \mathcal{S}^\perp \\ s_h^\perp, & \text{otherwise} \end{cases} p; a_h' = \begin{cases} a_h, & s_{1:h} \in \mathcal{S}^\perp \\ a^\perp, & \text{otherwise} \end{cases}; \ell_t'(s_h', a_h') = \begin{cases} \ell_t(s_h, a_h), & s_{1:h} \in \mathcal{S}^\perp \\ 0, & \text{otherwise} \end{cases}$$

12:        Send the pruned trajectory $o_t^\perp$ to `ALG`
13:    **end if**
14: **end for**

---

## 6 Improved regret bound for State-free RL

In the previous section, we introduce a black-box framework `SF-RL` that transforms any existing RL algorithm into a state-free RL algorithm. However, the regret guarantee for `SF-RL` is suboptimal: compared to `ALG` itself, `SF-RL` incurs an $\sqrt{|\mathcal{S}^{\Pi,\epsilon}|}$ multiplicative term to the regret bound. This is mainly because `SF-RL` needs to restart the black-box algorithm `ALG` whenever $\mathcal{S}^\perp$ updates. Such a restarting strategy inevitably leads to the loss of the learned MDP model. For this reason, and in order to achieve optimal regret rates, we need to design state-free algorithms that do not lose model information. In this section, we introduce a novel approach that enables `SF-RL` to retain previous transition information after restarting `ALG`. We illustrate that such a method improves the regret guarantee of `SF-RL` by a $\sqrt{|\mathcal{S}^{\Pi,\epsilon}|}$ term for adversarial MDPs when combined with a specific choice of `ALG`. This bound matches the best known regret bound for adversarial MDPs given known state space.

In existing adversarial MDP algorithms, the model information is captured within the confidence set of transition functions. Take Jin et al. [2019] as an example. For epoch $t \geq 1$, let $N_t(s, a)$ and $M_t(s'|s, a)$ be the total number of visits of pair $(s, a)$ and $(s, a, s')$ before epoch $t$. The confidence set of Jin et al. [2019] is defined as

$$\mathcal{P}_t = \left\{ \hat{P} : \left| \hat{P}(s'|s, a) - \bar{P}_t(s'|s, a) \right| \leq \epsilon_t(s'|s, a), \, \forall (s, a, s') \in \mathcal{S}_h \times \mathcal{A} \times \mathcal{S}_{h+1}, \, \forall h \right\},$$

where $\bar{P}_t(s'|s, a) = M_t(s'|s, a)/\max(1, N_t(s, a))$ is the empirical transition function for epoch $t$ and $\epsilon_t(s'|s, a)$ is the confidence width defined as

$$\epsilon_t(s'|s, a) = 2\sqrt{\frac{\bar{P}_t(s, a) \ln\left(\frac{4T|\mathcal{S}||\mathcal{A}|}{\delta}\right)}{\max\{1, N_t(s, a) - 1\}}} + \frac{14 \ln\left(\frac{4T|\mathcal{S}||\mathcal{A}|}{\delta}\right)}{3 \max\{1, N_t(s, a) - 1\}}.$$

As in Lemma 2 of Jin et al. [2019], by empirical Bernstein inequality and a union bound, one can establish that $P \in \mathcal{P}_t$ for all $t > 0$ with probability at least $1 - \delta$.

Intuitively, such a construction of confidence set tends to be overly conservative for our state-free setup. On the one hand, it requires taking a union bound over all $(s, a, s') \in \mathcal{S} \times \mathcal{A} \times \mathcal{S}$ pairs, resulting in an inevitable log-dependence on $|\mathcal{S}|$. On the other hand, even if state $s'$ is unreachable, the confidence width $\epsilon_t(s'|s, a)$ is not zero for all $(s, a)$. Furthermore, since `SF-RL` operates within the pruned space

$\mathcal{S}^\perp$, it is necessary to construct the confidence set on $S^\perp$ instead of $S$. Given by these observations, we propose a new construction of the confidence sets. For every $s \in \mathcal{S}$, we denote $t(s)$ by the epoch index when the algorithm first accesses to state $s$. If a state $s$ has not been visited, we define $t(s) = \infty$. Without of loss generality, we denote by $t(s,s') = \max\{t(s), t(s')\}$ the index when both $s$ and $s'$ are reached. Denote $\bar{P}_t^{t'}(s'|s,a) = (M_t(s'|s,a) - M_{t'}(s'|s,a))/\max\{1, N_t(s,a) - N_{t'}(s,a)\}$ be the partial empirical transition function corresponding to epochs $[t'+1, t]$. We further define $\mathcal{S}_t^\Pi$ as the states visited before $t$. For every $t \in [T]$, we build $\mathcal{P}_t^\perp$ such that

$$\mathcal{P}_t^\perp = \left\{ \begin{array}{ll} \hat{P}^\perp : \hat{P}^\perp(s'|s,a) \in \mathcal{I}_t(s'|s,a), & \forall (s,a,s') \in \mathcal{S}_h^\perp \setminus \{s_h^\perp\} \times \mathcal{A} \times \mathcal{S}_{h+1}^\perp \setminus \{s_{h+1}^\perp\}, \forall h \\ \hat{P}^\perp(s'|s,a) = \mathbb{1}\{s' = s_{h+1}^\perp\}, & \forall (s,a,s') \in \{s_h^\perp\} \times \mathcal{A} \times \mathcal{S}_{h+1}^\perp, \forall h \end{array} \right\}$$

where $\mathcal{I}_t(s'|s,a) = \mathcal{I}_t^1(s'|s,a) \cap \mathcal{I}_t^2(s'|s,a)$. $\mathcal{I}_t^1(s'|s,a)$ and $\mathcal{I}_t^2(s'|s,a)$ are two confidence intervals defined by

$$\mathcal{I}_t^1(s'|s,a) = \left[ \bar{P}_t^{t(s,s')}(s'|s,a) \pm \epsilon_t^1(s'|s,a) \right], \quad \mathcal{I}_t^2(s'|s,a) = \left\{ \begin{array}{ll} [0, \epsilon_t^2(s'|s,a)] & t(s') \geq t(s) + 1 \\ [0,1] & \text{else} \end{array} \right. ,$$

$$\epsilon_t^1(s'|s,a) = 4\sqrt{\frac{\bar{P}_t^{t(s,s')}(s'|s,a) \log\left(t/\delta(s,a,s')\right)}{\max\left\{N_t(s,a) - N_{t(s,s')}(s,a) - 1, 1\right\}}} + \frac{20\log\left(t/\delta(s,a,s')\right)}{\max\left\{N_t(s,a) - N_{t(s,s')}(s,a) - 1, 1\right\}},$$

$$\epsilon_t^2(s'|s,a) = \frac{2|\mathcal{S}_{t(s')}^\Pi| + 24\log\left(t/\delta(s,a)\right)}{\max\{N_{t(s')}(s,a) - 1, 1\}},$$

Here, $\mathcal{I}_t^1(s'|s,a)$ and $\mathcal{I}_t^2(s'|s,a)$ are two Bernstein-type confidence intervals. Let us explain the high-level ideas of the design. First, to avoid wasting confidence on unreachable states, we initialize the confidence level of $\mathcal{I}_t^1(s'|s,a)$ only if both $s$ and $s'$ are visited. The probability parameter $\delta(s'|s,a)$ is $\mathcal{F}_{t(s,s')}$-measurable, because it only depends on the data before epoch $t(s,s')$. Thus, in order to avoid correlation, we can only use the data collected after epoch $t(s,s') + 1$ to construct the confidence interval $\mathcal{I}_t^1(s'|s,a)$. This leads to a problem: when $t(s')$ is much greater than $t(s)$, we drop too much data that could be used to estimate $P(s'|s,a)$, resulting in $\mathcal{I}_t^1(s'|s,a)$ being loose compared to the existing confidence interval designed in Jin et al. [2019]. To address this issue, we introduce the second confidence interval $\mathcal{I}_t^2(s'|s,a)$. The logic behind the estimator $\mathcal{I}_t^2(s'|s,a)$ is that $t(s') \gg t(s)$ when the probability $P(s'|s,a)$ is very small and therefore $N_{t(s')}(s,a) - 1$ can be used to certify an upper bound to $P(s'|s,a)$. The confidence level of $\mathcal{I}_t^1(s'|s,a)$ can only be determined after epoch $t(s')$, whereas the confidence interval $\mathcal{I}_t^2(s'|s,a)$ is constructed based on data between $t(s)$ and $t(s')$. By combining $\mathcal{I}_t^1(s'|s,a)$ and $\mathcal{I}_t^2(s'|s,a)$, such a construction makes use of all the data after $t(s)$ to ensure a tight confidence interval. Considering that $t(s)$ is the first time $s$ is reached, we essentially lose only one data point, which is acceptable. By carefully designing the confidence level $\delta(s,a,s')$ and $\delta(s,a)$, one can show that

**Lemma 6.1.** *Let $i(s)$ be the index of state $s$ sorted by the arriving time. By setting $\delta(s,a) = \frac{\delta}{4i(s)^2|\mathcal{A}|}$ and $\delta(s,a,s') = \frac{\delta}{4(i(s)^4 + i(s')^4)|\mathcal{A}|}$, with probability at least $1 - \delta$, there is $P_t^\perp \in \mathcal{P}_t^\perp$ for all $t \in [T]$.*

Lemma 6.1 shows that such a construction of the confidence set is valid. Based on the new confidence set, we show that the regret bound of `SF-RL` can be improved by taking $\mathcal{P}_t^\perp$ as an additional input to the black-box algorithm `ALG`. We summarize the result as follows.

**Theorem 6.2.** *(Informal) By initializing `ALG` as `UOB-REPS` Jin et al. [2019] and taking $\mathcal{P}_t^\perp$ as an additional input to `ALG` every epoch, with probability $1 - \delta$, the state-free algorithm `SF-RL` achieves regret bound*

$$\mathbb{R}(T) \leq \mathcal{O}\left( H|\mathcal{S}^{\Pi,\epsilon}| \sqrt{|\mathcal{A}|T \log\left(\frac{|\mathcal{S}^\Pi||\mathcal{A}|T}{\delta}\right)} + \epsilon H |\mathcal{S}^\Pi| T \right),$$

*which matches the best existing result of non-state-free algorithms for adversarial MDPs.*

## 7 Conclusion

This paper initiates the study of state-free RL, where the algorithm does not require the information of state space as a hyper-parameter input. Our framework `SF-RL` allows us to transform any existing RL algorithm into a state-free RL algorithm through a black-box reduction. Future work includes extending the framework `SF-RL` from the tabular setting to the setting with function approximation.

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

# A    Omitted details for Section 2

## A.1    Details for Proposition 4.1

In this subsection, we illustrate that `UCBVI` is actually a weakly state-free algorithm. As in Azar et al. [2017], `UCBVI` algorithm consists of two parts: value iteration and policy execution. In every epoch, policy execution executes a policy that is greedy on the current $Q$ values and adds newly encountered trajectory to the dataset. Then, value iteration uses this dataset to update the $Q$ values of state-action pairs. Specifically, value iteration proceeds from the horizons $H, \ldots, 1$. For every $(s, a) \in \mathcal{S}_h \times \mathcal{A}$, the update can be expressed as

$$Q(s, a) = \min\{H, r(s, a) + \langle \bar{P}(\cdot|s, a), V(\cdot) \rangle + b(s, a)\},$$

where $\bar{P}(\cdot|s, a)$ is the empirical transitions estimation, $V(\cdot) = \max_{a \in \mathcal{A}} Q(\cdot, a)$ is the corresponding $V$ value, and $b(s, a)$ is the exploration bonus, which is defined by

$$b(s, a) = cHL\sqrt{\frac{1}{N_t(s, a)}}, \text{ where } L = \log\left(\frac{|\mathcal{S}||\mathcal{A}|T}{\delta}\right).$$

Throughout the algorithm, we can note that the knowledge of $\mathcal{S}$ is only applied in the design of the exploration bonus. Specifically, the exploration bonus is designed to ensure that the following event holds for all epochs with probability at least $1 - \delta$ by Hoeffding's inequality and a union bound.

$$\xi = \left\{ |\langle \bar{P}(\cdot|s, a) - P(\cdot|s, a), V^\star(\cdot) \rangle| \leq b(s, a), \ \forall (s, a) \in \mathcal{S} \times \mathcal{A} \right\} \tag{1}$$

When the reachable space $\mathcal{S}^\Pi$ is known, by substituting $S$ with $S^\Pi$ in advance, as in Theorem 2 of Azar et al. [2017], the regret of `UCBVI` is well bounded by $\mathcal{O}(H\sqrt{|\mathcal{S}^\Pi||\mathcal{A}|T \log(|\mathcal{S}^\Pi||\mathcal{A}|T/\delta)})$. When $\mathcal{S}^\Pi$ is unknown, we have to utilize $\mathcal{S}$ to design the bonus, resulting in the exploration bonus being amplified by a factor of $\sqrt{\log(|\mathcal{S}|)/\log(|\mathcal{S}^\pi|)}$. In this case, by optimism lemma, the regret guarantee increases by at most the same factor. Such a result suggests that `UCBVI` is weakly state-free.

## A.2    Details for Remark 4.2

Based on Proposition 4.1, here we show how to escape the log-dependence on $|\mathcal{S}|$ under the `UCBVI` framework. The idea is simple: when constructing the exploration bonus, instead of allocating confidence $\delta/|\mathcal{S}||\mathcal{A}|T$ to every state-action-epoch pair uniformly, we initialize the confidence level for states based on their arriving time. Let $i(s)$ be the index of state $s$ sorted by the arriving time. The exploration bonus is set by

$$b(s, a) = c'HL\sqrt{\frac{1}{\max\{N_t(s, a) - 1, 1\}}}, \text{ where } L = \log\left(\frac{2|i(s)|^2|\mathcal{A}|T}{\delta}\right).$$

Broadly speaking, for every visited state $s \in \mathcal{S}$, we allocate confidence $\delta/2|i(s)|^2|\mathcal{A}|T$ to its corresponding state-action-epoch pair. In this regard, the confidence allocated to state $s$ is bounded by $\delta/2|i(s)|^2$, and the total confidence is bounded by $\sum_{i=1}^\infty \delta/2i^2 \leq \delta$. Specifically, to avoid the correlation between the confidence level and the subsequent confidence sequence, we initialize the confidence sequence of $\langle \bar{P}(\cdot|s, a) - P(\cdot|s, a), V^\star(\cdot) \rangle$ at epoch $t(s) + 1$, where $t(s)$ is the epoch index that state $s$ is visited for the first time. This is because the confidence level $\delta/2|i(s)|^2|\mathcal{A}|T$ can be determined when the algorithm first reaches state $s$. In this way, we loose one data point for constructing the confidence sequence. This is why the count of visits is $\max\{N_t(s, a) - 1, 1\}$ rather than $N_t(s, a)$. Additionally, by the definition of $b(s, a)$, it suffices to note that $b(s, a) \geq H$ for the epochs before $t(s)$, which implies that the bonus ensures optimism for all epochs before $t(s)$ with probability 1. Combining with the above, it suffices to note that with probability $1 - \delta/2|i(s)|^2|\mathcal{A}|T$, the bonus $b(s, a)$ ensures optimism for all epochs. By a union bound, we can conclude that our proposed exploration bonuses also make the event in (1) hold with probability $1 - \delta$. Given that $i(s) \leq |\mathcal{S}^\Pi|$ for all visited states, the new exploration bonus is of the same order as the bonus where $\mathcal{S}^\Pi$ is known. In this way, we make the regret bound be completely independent to $|\mathcal{S}|$, and turn `UCBVI` to a fully state-free algorithm.

## A.3 Details for Proposition 4.3

In this subsection, we demonstrate that, within the current analysis framework for adversarial MDPs, eliminating the polynomial dependence on $|\mathcal{S}|$ is impossible. To the best of our knowledge, in order to handle the unknown transitions, all existing works share the same idea that maintain a confidence set $\mathcal{P}_t$ of transition functions for $t \in [T]$. The confidence set can be denoted by

$$\mathcal{P}_t \triangleq \left\{ \hat{P} : |\hat{P}(s'|s,a) - \bar{P}(s'|s,a)| \leq \epsilon_t(s'|s,a), \ \forall (s,a,s') \in \mathcal{S} \times \mathcal{A} \times \mathcal{S} \right\},$$

where $\epsilon_t(s'|s,a)$ is the confidence width and $\bar{P}(s'|s,a)$ is the empirical estimation of the true transition $P(s'|s,a)$. Specifically, $\epsilon_t(s'|s,a)$ is set based on empirical Bernstein inequality, i.e.,

$$\epsilon_t(s'|s,a) = \mathcal{O}\left( \sqrt{\frac{\bar{P}(s'|s,a)\log(SAT/\delta)}{N_t(s,a)}} + \frac{\log(SAT/\delta)}{N_t(s,a)} \right), \ \forall (s,a,s') \in \mathcal{S} \times \mathcal{A} \times \mathcal{S},$$

where $N_t(s,a)$ denotes the counter of state-action pair $(s,a)$. This setting ensures that $P \in \mathcal{P}_t$ for all $t \in [T]$ with probability at least $1 - \delta$. Given the confidence set, the algorithm selects $\hat{P}_t \in \mathcal{P}_t$ as the approximation of $P$, and chooses policy $\pi_t$ based on the approximation transition. For a clear understanding, we decompose the regret into two terms, i.e.,

$$\mathbb{R}(T) = \sum_{t=1}^T \langle q^{P,\pi_t} - q^{P,\pi^*}, \ell_t \rangle = \underbrace{\sum_{t=1}^T \langle q^{\hat{P}_t,\pi_t} - q^{P,\pi^*}, \ell_t \rangle}_{\text{REGRET}} + \underbrace{\sum_{t=1}^T \langle q^{P,\pi_t} - q^{\hat{P}_t,\pi_t}, \ell_t \rangle}_{\text{ERROR}}$$

Here, the first term REGRET represents the regret of the algorithm with the approximation transition. In some senses, bounding REGRET can be reduced to a bandit problem. In every round $t$, the algorithm chooses an occupancy measure $\hat{q}_t \in \Delta(\mathcal{P}_t, \Pi)$ and corresponding $(\hat{P}_t, \pi_t) \in (\mathcal{P}_t, \Pi)$, then obtain a partial observation of the loss $\ell_t$. For the second term ERROR, it corresponds to the error using $\hat{P}_t$ to approximate $P$. Considering the adversarial environment, there exists a "worst enough" loss sequence $\ell_1, \ldots, \ell_T$ such that

$$\text{ERROR} \approx \sum_{t=1}^T \sum_{(s,a) \in \mathcal{S}^\Pi \times \mathcal{A}} |q^{P,\pi_t}(s,a) - q^{\hat{P}_t,\pi_t}(s,a)|.$$

Thus, bounding ERROR is essentially equates to bounding the right hand side of the above. At this point, one needs to demonstrate that the confidence set shrinks in the correct rate over time, so that the sum of the gap between $q^{P,\pi_t}$ and $q^{\hat{P}_t,\pi_t}$ can be well bounded.

Having provided sufficient background, we now explain why existing methods fail to achieve state-free regret bounds. First, since the confidence set requires to work for all $(s,a,s')$ pairs, we have to take a union bound on the "good event" for all $(s,a,s') \in \mathcal{S} \times \mathcal{A} \times \mathcal{S}$. This essentially cause $\epsilon_t(s'|s,a)$ to be log-dependent on $|\mathcal{S}|$. More important, to bound $|q^{P,\pi_t}(s,a) - q^{\hat{P}_t,\pi_t}(s,a)|$, existing methods mainly follow the proof of Lemma 4 in Jin et al. [2019], that is, for every $(s,a) \in \mathcal{S}^\Pi \times \mathcal{A}$ and $\pi_t \in \Pi$, there exists $\hat{P}_t \in \mathcal{P}_t$ such that

$$\left| q^{\hat{P}_t,\pi_t}(s,a) - q^{P,\pi_t}(s,a) \right| \approx \sum_{h=1}^{h(s)-1} \sum_{s_h,a_h,s_{h+1}} \epsilon_t(s_{h+1}|s_h,a_h) q^{P,\pi_t}(s_h,a_h) q^{\hat{P}_t,\pi_t}(s,a|s_{h+1}).$$

By the definition of confidence width, we have $\epsilon_t(s_{h+1}|s_h,a_h) \geq \tilde{\mathcal{O}}(1/N_t(s_h,a_h))$ for all $(s_h, a_h, s_{h+1})$ pairs. Furthermore, if state $s_{h+1}$ is unvisited, the algorithm has no information for the transitions after the state. Assuming that $\mathcal{S}_h \neq \mathcal{S}_h^\Pi$ for all $h \in [H]$, there will always exist a "worst enough" $\hat{P}_t \in \mathcal{P}_t$ such that the probability of reaching $s$ via $s_{h+1}$ with policy $\pi_t$ is 1, i.e., $q^{\hat{P}_t,\pi_t}(s|s_{h+1}) = 1$. In this case, we have

$$\sum_{a \in \mathcal{A}} \left| q^{\hat{P}_t,\pi_t}(s,a) - q^{P,\pi_t}(s,a) \right|$$

$$\approx \sum_{h=1}^{h(s)-1} \sum_{s_h,a_h,s_{h+1}} \tilde{\mathcal{O}}\left(\frac{1}{N_t(s_h,a_h)}\right) q^{P,\pi_t}(s_h,a_h) \sum_{a\in\mathcal{A}} q^{\hat{P}_t,\pi_t}(s,a|s_{h+1})$$

$$\geq \sum_{h=1}^{h(s)-1} \sum_{s_h,a_h,s_{h+1}} \tilde{\mathcal{O}}\left(\frac{1}{N_t(s_h,a_h)}\right) q^{P,\pi_t}(s_h,a_h) \mathbb{1}\{s_{h+1} \text{ is unreachable}\}$$

$$\geq \tilde{\mathcal{O}}\left(\frac{1}{t}\right) \sum_{h=1}^{h(s)-1} \sum_{s_h,a_h} q^{P,\pi_t}(s_h,a_h)(|\mathcal{S}_{h+1}| - |\mathcal{S}_{h+1}^{\Pi}|)$$

$$= \tilde{\mathcal{O}}\left(\frac{\sum_{h=1}^{h(s)-1}(|\mathcal{S}_{h+1}| - |\mathcal{S}_{h+1}^{\Pi}|)}{t}\right).$$

Based on the analysis, it suffices to see that ERROR is at least of order $\tilde{\mathcal{O}}(\sum_{m\in[H]} |\mathcal{S}_h^{\Pi}| \sum_{h=1}^{h(s)-1}(|\mathcal{S}_{h+1}| - |\mathcal{S}_{h+1}^{\Pi}|))$, which polynomially depends on $|\mathcal{S}|$ when $|\mathcal{S}^{\Pi}| \ll |\mathcal{S}|$. Such a result suggests that current analysis cannot derive state-free regret bound for adversarial MDP.

# B Omitted proof of Section 3

## B.1 Proof of Lemma 5.3

Fix $S^{\perp}$, we consider an pruned trajectory $o_t^{\perp}$ derived by SF-RL such that

$$o_t^{\perp} = \{s_1, a_1, \ell_t(s_1, a_1), \ldots, s_h, a_h, \ell_t(s_h, a_h), s_{h+1}^{\perp}, a^{\perp}, 0, \ldots, s_H^{\perp}, a^{\perp}, 0\}.$$

By definition, it suffices to note that $s_{1:h} \in \mathcal{S}^{\perp}$ and and $s_{h+1} \notin \mathcal{S}^{\perp}$. In the following, we complete the proof by demonstrating that the likelihood of obtaining $o_t^{\perp}$ using algorithm SF-RL is the same to the likelihood of obtaining $o_t^{\perp}$ by executing $\pi_t^{\perp}$ on $\mathcal{P}^{\perp}$ and $\ell_t^{\perp}$.

We first analyze the likelihood of obtaining $o_t^{\perp}$ by SF-RL. Let's first review the algorithm. Initially, SF-RL executes policy $\pi_t$ and obtains the trajectory $o_t$, which is on the underlying transition $P$ and loss $\ell_t$. Then, using $\mathcal{S}^{\perp}$, SF-RL degrades $o_t$ to the pruned trajectory $o_t^{\perp}$. In order to generate $o_t^{\perp}$ defined above, $o_t$ needs to satisfy 1). at horizon 1 to $h$, the state-action pairs are $(s_1, a_1)$ to $(s_h, a_h)$ respectively. 2). at horizon $h+1$, the visited state cannot belong to $\mathcal{S}^{\perp}$. Therefore, the likelihood can be denoted by

$$\mathbb{P}(o_t^{\perp}|\text{SF-RL}) = \underbrace{\pi_t(a_h|s_h) \prod_{k=0}^{h-1} \pi_t(a_k|s_k) \prod_{k=0}^{h-1} P(s_{k+1}|s_k, a_k)}_{\text{Likelihood of visiting } \{s_k,a_k\}_{k=1}^h} \underbrace{\left(1 - \sum_{s^{\dagger}\in\mathcal{S}_{h+1}^{\perp}\backslash\{s_{h+1}^{\perp}\}} P(s^{\dagger}|s,a)\right)}_{\text{Likelihood of visiting states not in } \mathcal{S}^{\perp} \text{ at horizon } h+1}$$

Now we study the likelihood of obtaining $o_t^{\perp}$ by executing $\pi_t^{\perp}$ on $\mathcal{P}^{\perp}$ and $\ell_t^{\perp}$. By definition, for every $s \in \mathcal{S}$, there is $\ell_t^{\perp}(s, a) = \ell_t(s, a)$ if $s \in \mathcal{S}^{\perp}$, and $\ell_t^{\perp}(s, a) = 0$ otherwise. Using the observation, we can rewrite the loss $\ell_t(s_k, a_k)$ by $\ell_t^{\perp}(s_k, a_k)$ for $k \in 1, \ldots, h$, and rewrite the rest zero loss by $\ell_t^{\perp}(s_k^{\perp}, a^{\perp})$. Therefore, it suffices to focus on the likelihood of obtaining state-action pairs of $o_t^{\perp}$. In this regard, we have

$\mathbb{P}(o_t^{\perp}|P^{\perp}, \ell_t^{\perp}, \pi_t^{\perp})$

$$= \left(\prod_{k=0}^{h-1} \pi_t^{\perp}(a_k|s_k) \prod_{k=0}^{h-1} P^{\perp}(s_{k+1}|s_k, a_k)\right) \pi_t^{\perp}(a_h|s_h) P^{\perp}(s_{h+1}^{\perp}|s_h, a_h) \left(\prod_{k=h+1}^{H} \pi_t^{\perp}(a^{\perp}|s_k) \prod_{k=0}^{h-1} P^{\perp}(s_{k+1}^{\perp}|s_k^{\perp}, a^{\perp})\right)$$

$$= \left(\prod_{k=0}^{h-1} \pi_t^{\perp}(a_k|s_k) \prod_{k=0}^{h-1} P^{\perp}(s_{k+1}|s_k, a_k)\right) \pi_t^{\perp}(a_h|s_h) P^{\perp}(s_{h+1}^{\perp}|s_h, a_h)$$

$$= \left(\prod_{k=0}^{h-1} \pi_t(a_k|s_k) \prod_{k=0}^{h-1} P(s_{k+1}|s_k, a_k)\right) \pi_t(a_h|s_h) P^{\perp}(s_{h+1}^{\perp}|s_h, a_h)$$

$$= \prod_{k=0}^{h} \pi_t(a_k|s_k) \prod_{k=0}^{h-1} P(s_{k+1}|s_k, a_k)\pi_t(a_h|s_h) \left(1 - \sum_{s^{\dagger}\in\mathcal{S}_{h+1}^{\perp}\backslash\{s_{h+1}^{\perp}\}} P(s^{\dagger}|s,a)\right)$$

where the first equality is because $\prod_{k=h+1}^{H} \pi_t^{\perp}(a^{\perp}|s_k)P^{\perp}(s_{k+1}^{\perp}|s_k^{\perp}, a^{\perp}) = 1$ by definition. The second and third equalities are by the definition of $P^{\perp}$ and $\pi_t^{\perp}$, that is, $\pi_t^{\perp}(a_k|s_k) = \pi_t(s_k, a_k)$ and $P^{\perp}(s_{k+1}|s_k, a_k) = P(s_{k+1}|s_k, a_k)$ for $k = 0, \ldots, h$, The last equality is by the definition of $P^{\perp}(s_{h+1}^{\perp}|s_h, a_h)$. Using the above, it suffices to show that $\mathbb{P}(o_t^{\perp}|\text{SF-RL}) = \mathbb{P}(o_t^{\perp}|P^{\perp}, \ell_t^{\perp}, \pi_t^{\perp})$, thereby we complete the proof.

## B.2 Proof of Lemma 5.4

The proof is mainly based on the following lemma.

**Lemma B.1.** *Let $\mathcal{F}_t$ for $t > 0$ be a filtration and $(X_t \geq 0)_{t \in \mathbb{N}^+}$ be a sequence of non-negative random variables with $\mathbb{E}[X_t|\mathcal{F}_{t-1}] = P_t$ with $P_t$ being $\mathcal{F}_{t-1}$-measurable. Given confidence level $\delta$ being $\mathcal{F}_t$-measurable, there is*

$$\mathbb{P}\left(\exists n > t, \sum_{j=t+1}^{n} X_j \geq 2\sum_{j=t+1}^{n} P_j + \log\frac{1}{\delta}\right) \leq \delta,$$

$$\mathbb{P}\left(\exists n > t, \sum_{j=t+1}^{n} P_j \geq 2\sum_{j=t+1}^{n} X_j + \log\frac{1}{\delta}\right) \leq \delta.$$

Let $i(s)$ be the index of state $s$ sorted by the arriving time. Recall $t(s)$ is the epoch when the algorithm first accesses to state $s$. Apparently, $i(s)$ is $\mathcal{F}_{t(s)}$-measurable. Using Lemma B.1 and a union bound, we immediately have

$$\mathbb{P}\left(\forall s \in \mathcal{S}, \forall n > t(s), \sum_{j=t(s)+1}^{n} q^{P,\pi_j}(s) > \sum_{j=t(s)+1}^{n} \frac{\mathbb{1}_j\{s\}}{2} - \frac{\log(2i(s)^2/\delta)}{2}\right) \geq 1 - \delta.$$

Assuming the above holds for true. By SF-RL, a state $s$ will be added in $\mathcal{S}^{\perp}$ only if $\sum_{j=1}^{t} \mathbb{1}_j\{s\}/2 - \log(2i(s)^2/\delta)/2 - 1/2 \geq \epsilon t$. Since there are at most $Hn$ states that can be visited before epoch $t$, we have $i(s) \leq Ht(s) \leq Hn$. Moreover, it is obvious that $\sum_{j=1}^{t} \mathbb{1}_j\{s\} = \sum_{j=t(s)+1}^{t} \mathbb{1}_j\{s\} + 1$ by the definition of $t(s)$. Thus, we have

$$n \max_{\pi \in \Pi} q^{P,\pi}(s) \geq \sum_{j=t(s)+1}^{n} q^{P,\pi_j}(s) > \sum_{j=t(s)+1}^{n} \frac{\mathbb{1}_j\{s\}}{2} - \frac{\log(2i(s)^2/\delta)}{2}$$

$$= \sum_{j=1}^{n} \frac{\mathbb{1}_j\{s\}}{2} - \frac{\log(2H^2n^2/\delta)}{2} - \frac{1}{2} \geq \epsilon n,$$

which implies that state $s$ is $\epsilon$-reachable. This completes the proof.

## B.3 Proof of Lemma 5.5

To prove Lemma 5.5, the key observation is that

$$\langle q^{P^{\perp},\pi^{\perp}}, \ell_t^{\perp}\rangle = \mathbb{E}\left[\sum_{h=1}^{H} \ell_t^{\perp}(s_h, a_h)|P^{\perp}, \pi^{\perp}\right] = \mathbb{E}\left[\sum_{h=1}^{H} \ell_t(s_h, a_h)\mathbb{1}\{s_{1:h} \in \mathcal{S}^{\perp}\}|P, \pi\right],$$

where the first equality is by occupancy measure and the second equality is by Lemma 5.3. Using the observation, we have

$$\langle q^{P,\pi}, \ell_t\rangle - \langle q^{P^{\perp},\pi^{\perp}}, \ell_t^{\perp}\rangle = \mathbb{E}\left[\sum_{h=1}^{H} \ell_t(s_h, a_h)|P, \pi\right] - \mathbb{E}\left[\sum_{h=1}^{H} \ell_t^{\perp}(s_h, a_h)|P^{\perp}, \pi^{\perp}\right]$$

$$= \mathbb{E}\left[\sum_{h=1}^{H} \ell_t(s_h, a_h)|P, \pi\right] - \mathbb{E}\left[\sum_{h=1}^{H} \ell_t(s_h, a_h)\mathbb{1}\{s_{1:h} \in \mathcal{S}^{\perp}\}|P, \pi\right],$$

thus we have $\langle q^{P,\pi}, \ell_t\rangle - \langle q^{P^\perp,\pi^\perp}, \ell_t^\perp\rangle \ge 0$ and

$$\langle q^{P,\pi}, \ell_t\rangle - \langle q^{P^\perp,\pi^\perp}, \ell_t^\perp\rangle \le H\mathbb{E}\left[\mathbb{1}\{\exists h, s_h \notin \mathcal{S}^\perp\}|P,\pi\right] \le H\sum_{s\in\mathcal{S}^\Pi} q^{P,\pi}(s)\mathbb{1}\{s\notin\mathcal{S}^\perp\},$$

which completes the proof.

# C Proof of Section 4

## C.1 Proof of Lemma 6.1

We first fix a horizon $h$. For every $(s,a,s') \in \mathcal{S}_h \times \mathcal{A} \times \mathcal{S}_{h+1}$, considering that $\delta(s,a,s')$ is $\mathcal{F}_{t(s,s')}$-measurable, by empirical Bernstein inequality and a union bound, we immediately have $P(s'|s,a) \in \mathcal{I}_t^1(s'|s,a)$ for all $t \ge t(s,s')+1$ with probability $1 - \delta(s,a,s')$. By the definition of $\mathcal{P}^\perp$, there is $P^\perp(s'|s,a) = P(s'|s,a)$ once $s,s' \in \mathcal{S}^\perp$. Furthermore, the confidence sequence of $P^\perp(s'|s,a)$ is initialized once both states $s$ and $s'$ have been visited, which is potentially as soon as epoch $t(s,s')+1$. Given $\sum_{s\in\mathcal{S},a\in\mathcal{A},s'\in\mathcal{S}} \delta(s,a,s') \le \delta/2$, it suffices to say the following event

$$\xi_1 = \left\{P^\perp(s'|s,a) \in \mathcal{I}_t^1(s'|s,a); \forall t \in [T], (s,a,s') \in \mathcal{S}_h^\perp \setminus \{s_h^\perp\} \times \mathcal{A} \times \mathcal{S}_{h+1}^\perp \setminus \{s_{h+1}^\perp\}, h\right\}$$

holds true with probability at least $1 - \delta/2$.

It now suffices to focus on the second confidence interval $\mathcal{I}_t^2(s'|s,a)$. For every $(s,a) \in \mathcal{S}_h \times \mathcal{A}$, we define

$$\mathcal{S}_t^{s,a} = \left\{s' \in \mathcal{S}| \sum_{\tau=t(s)}^{t-1} \mathbb{1}_\tau\{s',s,a\} = 0\right\}$$

be the states such that the state-action-state pair $(s,a,s')$ is unvisited before epoch $t$. Notice that $\mathcal{S}_t^{s,a}$ is $\mathcal{F}_{t-1}$-measurable and $\mathbb{E}[\mathbb{1}_t\{\mathcal{S}_t^{s,a}|s,a\}|\mathcal{F}_{t-1}] = P(\mathcal{S}_t^{s,a}|s,a)$. Given a $\mathcal{F}_{t(s)}$-measurable confidence $\delta(s,a)$, by empirical Bernstein inequality and a union bound, it suffices to claim that

$$\left|\sum_{\tau=t(s)+1}^{t} P(\mathcal{S}_\tau^{s,a}|s,a)\mathbb{1}_\tau\{s,a\} - \sum_{\tau=t(s)+1}^{t} \mathbb{1}_\tau\{\mathcal{S}_\tau^{s,a}|s,a\}\mathbb{1}_\tau\{s,a\}\right|$$

$$\le \sqrt{2\sum_{\tau=t(s)+1}^{t} \mathbb{1}_\tau\{\mathcal{S}_\tau^{s,a}|s,a\}\mathbb{1}_\tau\{s,a\}\log\left(\frac{2t^2}{\delta(s,a)}\right)} + \frac{14\log\left(\frac{2t^2}{\delta(s,a)}\right)}{3}, \quad \forall t \ge t(s)+1$$

with probability at least $1 - \delta(s,a)$.

By the definition of $\mathcal{S}_t^{s,a}$, it suffices to note that $\sum_{\tau=t(s)+1}^{t} \mathbb{1}_\tau\{\mathcal{S}_\tau^{s,a}|s,a\}\mathbb{1}_\tau\{s,a\} \le |\mathcal{S}_t^\Pi|$. Using the above, we can note that

$$\sum_{\tau=t(s)+1}^{t} P(\mathcal{S}_\tau^{s,a}|s,a)\mathbb{1}_\tau\{s,a\} \le |\mathcal{S}_t^\Pi| + \sqrt{2|\mathcal{S}_t^\Pi|\log\left(\frac{2t^2}{\delta(s,a)}\right)} + \frac{14\log\left(\frac{2t^2}{\delta(s,a)}\right)}{3}$$

$$\le 2|\mathcal{S}_t^\Pi| + 24\log\left(\frac{t}{\delta(s,a)}\right)$$

with probability at least $1 - \delta(s,a)$.

Consider a state $s' \in \mathcal{S}$ such that $t(s') \ge t(s)+1$. By definition, it suffices to note that $s' \in \mathcal{S}_\tau^{s,a}$ for all $t(s)+1 \le \tau \le t(s')$, which implies that $P(s'|s,a) \le P(\mathcal{S}_\tau^{s,a}|s,a)$ for all $t(s)+1 \le \tau \le t(s')$. In this case, we finally have

$$P(s'|s,a) \le \frac{\sum_{\tau=t(s)+1}^{t(s')} P(\mathcal{S}_\tau^{s,a}|s,a)\mathbb{1}_\tau\{s,a\}}{\sum_{\tau=t(s)+1}^{t(s')} \mathbb{1}_\tau\{s,a\}} \le \frac{2|\mathcal{S}_{t(s')}^\Pi| + 24\log\left(\frac{t(s')}{\delta(s,a)}\right)}{\max\{N_{t(s')}(s,a)-1,1\}}, \quad \forall s' : t(s') \ge t(s)+1$$

with probability at least $1 - \delta(s,a)$. Given $\sum_{s \in \mathcal{S}, a \in \mathcal{A}} \delta(s,a) \leq \delta/2$ and $P^\perp(s'|s,a) = P(s'|s,a)$, it suffices to say that the following event

$$\xi_2 = \left\{ P^\perp(s'|s,a) \in \mathcal{I}_t^2(s'|s,a); \forall t \in [T], (s,a,s') \in \mathcal{S}_h^\perp \setminus \{s_h^\perp\} \times \mathcal{A} \times \mathcal{S}_{h+1}^\perp \setminus \{s_{h+1}^\perp\}, h \right\}$$

holds true with probability at least $1 - \delta/2$. By a union bound of events $\xi_1$ and $\xi_2$, we complete the proof.

## C.2 Proof of Theorem 6.2

The proof of Theorem 6.2 is technical but mostly follows the same ideas of that for Lemma 4 in Jin et al. [2019]. First, as in Jin et al. [2019], we illustrate that the confidence set $\mathcal{P}_t^\perp$ is tight enough, i.e., the difference between the true transition function and any transition function from the confidence set can be well bounded.

**Lemma C.1.** *Under the event of Lemma 6.1, for all epoch $t \in [T]$, all $\hat{P}_t^\perp \in \mathcal{P}_t^\perp$, all $h = 0, \ldots, H-1$ and $(s,a,s') \in \mathcal{S}_h^\perp \setminus \{s_h^\perp\} \times \mathcal{A} \times \mathcal{S}_{h+1}^\perp \setminus \{s_{h+1}^\perp\}$, we have*

$$\left| \hat{P}_t^\perp(s'|s,a) - P_t^\perp(s'|s,a) \right| \leq \mathcal{O} \left( \sqrt{\frac{P(s'|s,a) \log\left(\frac{|\mathcal{S}^\Pi||\mathcal{A}|T}{\delta}\right)}{\max\{N_t(s,a),1\}}} + \frac{|\mathcal{S}^\Pi| + \log\left(\frac{|\mathcal{S}^\Pi||\mathcal{A}|T}{\delta}\right)}{\max\{N_t(s,a),1\}} \right) \triangleq \epsilon_t^\star(s'|s,a)$$

**Lemma C.2.** *(Refined regret guarantee for Theorem 3 in Jin et al. [2019]) With probability $1 - \delta$, for all $K > 0$, with confidence sets $\mathcal{P}_1, \ldots, \mathcal{P}_K$, the regret guarantee for $\mathtt{UOB\text{-}REPS}$ following $K$ epochs of interaction with MDP $\mathcal{M} = (\mathcal{S}, \mathcal{A}, H, P)$ and loss sequence $\ell_1, \ldots, \ell_K$ is bounded by*

$$\mathbb{R}^{\mathtt{UOB\text{-}REPS}}(K) \leq \mathcal{O} \left( \sqrt{H|\mathcal{S}||\mathcal{A}|K \log(|\mathcal{S}||\mathcal{A}|K/\delta)} + \sum_{k=1}^{K} \sum_{s \in \mathcal{S}, a \in \mathcal{A}} \left| q^{P_k^s, \pi_k}(s,a) - q^{P, \pi_k}(s,a) \right| |\ell_k(s,a)| \right),$$

*where $\{\pi_k\}_{k \in [K]}$ is a collection of policies and $\{\hat{P}_k^s\}_{s \in \mathcal{S}, k \in [K]}$ is a collection of transition functions selected by pessimism, i.e., for all $s \in \mathcal{S}$ and $k \in [K]$,*

$$P_k^s = \arg\max_{\hat{P} \in \mathcal{P}_k} \sum_{a \in \mathcal{A}} \left| q^{\hat{P}, \pi_k}(s,a) - q^{P, \pi_k}(s,a) \right| |\ell_k(s,a)|.$$

Recall the proof sketch in Section 5, we decompose the regret $\mathbb{R}(T)$ into ① and ②, which represent $\mathtt{ALG}$'s regret and the error incurred by the difference between $\mathcal{S}$ and $\mathcal{S}^\perp$ respectively. By the proof of Theorem 5.2, there is ② $\leq \mathcal{O}(\epsilon H|\mathcal{S}^\Pi|T)$. It suffices to focus on ①. By Lemma C.2, we have

$$① \leq \mathcal{O} \left( \sum_{m=1}^{M} \sqrt{H|\mathcal{S}_{(m)}^\perp||\mathcal{A}||\mathcal{I}_m| \log(|\mathcal{S}_{(m)}^\perp||\mathcal{A}||\mathcal{I}_m|/\delta)} + \sum_{t=1}^{T} \sum_{s \in \mathcal{S}_t^\perp, a \in \mathcal{A}} \left| q^{P_t^s, \pi_t^\perp}(s,a) - q^{P_t^\perp, \pi_t^\perp}(s,a) \right| |\ell_t^\perp(s,a)| \right)$$

$$\leq \mathcal{O} \left( H|\mathcal{S}^{\Pi,\epsilon}| \sqrt{|\mathcal{A}|T \log(|\mathcal{S}^{\Pi,\epsilon}||\mathcal{A}|T/\delta)} + \sum_{t=1}^{T} \sum_{s \in \mathcal{S}_t^\perp \setminus \{s_h^\perp\}_{h \in [H]}, a \in \mathcal{A}} \left| q^{P_t^s, \pi_t^\perp}(s,a) - q^{P_t^\perp, \pi_t^\perp}(s,a) \right| \right),$$

where $P_t^s \in \mathcal{P}_t^\perp$ for all $t \in [T]$ and $s \in \mathcal{S}_t^\perp$. It suffices to focus on the second term. For every $s \in \mathcal{S}_t^\perp \setminus \{s_h^\perp\}_{h \in [H]}, a \in \mathcal{A}$, let $h(s)$ be the index of horizon to which $s$ belongs. According to the proof of Lemma 4 in Jin et al. [2019] (specifically their Eq. (15)), we have

$$\left| q^{P_t^s, \pi_t^\perp}(s,a) - q^{P_t^\perp, \pi_t^\perp}(s,a) \right|$$

$$\leq \sum_{m=0}^{h(s)-1} \sum_{s_m \in \mathcal{S}_{t,m}^\perp, a_m \in \mathcal{A}, s_{m+1} \in \mathcal{S}_{t,m+1}^\perp} \left| P_t^s(s_{m+1}|s_m,a_m) - P_t^\perp(s_{m+1}|s_m,a_m) \right| q^{P_t^\perp, \pi_t^\perp}(s_m,a_m) q^{P_t^s, \pi_t^\perp}(s,a|s_{m+1})$$

where $\mathcal{S}_{t,m}^\perp$ represents the states $s \in \mathcal{S}_t^\perp$ at horizon $m$. Intuitively, in order to continue the proof, we should apply Lemma C.1 and bound $|P_t^s(s_{m+1}|s_m,a_m) - P_t^\perp(s_{m+1}|s_m,a_m)|$ by $\epsilon_t^\star(s_{m+1}|s_m,a_m)$. However, when $s_m = s_m^\perp$ or $s_{m+1} = s_{m+1}^\perp$, the confidence width $\epsilon_t^\star(s_{m+1}|s_m,a_m)$ is not well defined. To address this, we show that the terms related to states $s_m^\perp$ or $s_{m+1}^\perp$ can be disregarded. We prove case by case, i.e.,

1. ($s_m = s_m^\perp$): By the definition of $P_t^\perp$ and $\mathcal{P}_t^\perp$, we always have
$$P_t^\perp(s_{m+1}|s_m, a_m) = P_t^s(s_{m+1}|s_m, a_m) = \mathbb{1}\{s_{m+1} = s_{m+1}^\perp\},$$
which implies that $|P_t^\perp(s_{m+1}|s_m, a_m) - P_t^s(s_{m+1}|s_m, a_m)| = 0$.

2. ($s_m \neq s_m^\perp, s_{m+1} = s_{m+1}^\perp$): By the definition of $\mathcal{P}_t^\perp$, after visiting state $s_{m+1}^\perp$, the probability of visiting state $s \neq s_{h(s)}^\perp$ is zero. This means that $q^{P_t^s, \pi_t^\perp}(s, a|s_{m+1}) = 0$.

3. ($s_m \neq s_m^\perp, s_{m+1} \neq s_{m+1}^\perp$): By Lemma C.1, we can bound $|P_t^s(s_{m+1}|s_m, a_m) - P_t^\perp(s_{m+1}|s_m, a_m)|$ by $\epsilon_t^\star(s_{m+1}|s_m, a_m)$.

Using the above, it suffices to show that

$$\left| q^{P_t^s, \pi_t^\perp}(s, a) - q^{P_t^\perp, \pi_t^\perp}(s, a) \right|$$

$$\leq \sum_{m=0}^{h(s)-1} \sum_{s_m \in \mathcal{S}_{t,m}^\perp \setminus \{s_m^\perp\}, a_m \in \mathcal{A}, s_{m+1} \in \mathcal{S}_{t,m+1}^\perp \setminus \{s_{m+1}^\perp\}} \epsilon_t^\star(s_{m+1}|s_m, a_m) q^{P_t^\perp, \pi_t^\perp}(s_m, a_m) q^{P_t^s, \pi_t^\perp}(s, a|s_{m+1})$$

$$\leq \sum_{m=0}^{h(s)-1} \sum_{s_m \in \mathcal{S}_{t,m}^\perp \setminus \{s_m^\perp\}, a_m \in \mathcal{A}, s_{m+1} \in \mathcal{S}_{t,m+1}^\perp \setminus \{s_{m+1}^\perp\}} \epsilon_t^\star(s_{m+1}|s_m, a_m) q^{P, \pi_t}(s_m, a_m) q^{P_t^s, \pi_t^\perp}(s, a|s_{m+1}),$$

where the second inequality is by Lemma 5.5, i.e.,

$$q^{P_t^\perp, \pi_t^\perp}(s_m, a_m) = \langle q^{P_t^\perp, \pi_t^\perp}, \mathbb{1}\{s_m, a_m\} \rangle \leq \langle q^{P, \pi_t}, \mathbb{1}\{s_m, a_m\} \rangle = q^{P, \pi_t}(s_m, a_m).$$

By the exact same analysis, we also have

$$\left| q^{P_t^s, \pi_t^\perp}(s, a|s_{m+1}) - q^{P_t^\perp, \pi_t^\perp}(s, a|s_{m+1}) \right|$$

$$\leq \sum_{h=m+1}^{h(s)-1} \sum_{s_h' \in \mathcal{S}_{t,h}^\perp \setminus \{s_h^\perp\}, a_h' \in \mathcal{A}, s_{h+1}' \in \mathcal{S}_{t,h+1}^\perp \setminus \{s_{h+1}^\perp\}} \epsilon_t^\star(s_{h+1}'|s_h', a_h') q^{P_t^\perp, \pi_t^\perp}(s_h', a_h'|s_{m+1}) q^{P_t^s, \pi_t^\perp}(s, a|s_{h+1}')$$

$$\leq \pi_t^\perp(a|s) \sum_{h=m+1}^{h(s)-1} \sum_{s_h' \in \mathcal{S}_{t,h}^\perp \setminus \{s_h^\perp\}, a_h' \in \mathcal{A}, s_{h+1}' \in \mathcal{S}_{t,h+1}^\perp \setminus \{s_{h+1}^\perp\}} \epsilon_t^\star(s_{h+1}'|s_h', a_h') q^{P_t^\perp, \pi_t^\perp}(s_h', a_h'|s_{m+1})$$

$$\leq \pi_t(a|s) \sum_{h=m+1}^{h(s)-1} \sum_{s_h' \in \mathcal{S}_{t,h}^\perp \setminus \{s_h^\perp\}, a_h' \in \mathcal{A}, s_{h+1}' \in \mathcal{S}_{t,h+1}^\perp \setminus \{s_{h+1}^\perp\}} \epsilon_t^\star(s_{h+1}'|s_h', a_h') q^{P, \pi}(s_h', a_h'|s_{m+1})$$

To simplify notation, in the following, we use the shorthands $w_h = (s_h, a_h, s_{h+1})$ and $\epsilon_t^\star(w_h) = \epsilon_t^\star(s_{h+1}|s_h, a_h)$. We further denote $W_{t,h}^\perp$ by all the state-action-state pairs $\mathcal{S}_{t,h}^\perp \setminus \{s_h^\perp\} \times \mathcal{A} \times \mathcal{S}_{t,h+1}^\perp \setminus \{s_{h+1}^\perp\}$ and $W_h^{\Pi,\epsilon}$ by all the reachable state-action-state pairs $\mathcal{S}_h^{\Pi,\epsilon} \times \mathcal{A} \times \mathcal{S}_{h+1}^{\Pi,\epsilon}$. Using the above two inequalities, we have

$$\left| q^{P_t^s, \pi_t^\perp}(s, a) - q^{P_t^\perp, \pi_t^\perp}(s, a) \right|$$

$$\leq \sum_{m=0}^{h(s)-1} \sum_{w_m \in W_{t,m}^\perp} \epsilon_t^\star(w_m) q^{P, \pi_t}(s_m, a_m) q^{P_t^\perp, \pi_t^\perp}(s, a|s_{m+1})$$

$$+ \sum_{m=0}^{h(s)-1} \sum_{w_m \in W_{t,m}^\perp} \epsilon_t^\star(w_m) q^{P_t^\perp, \pi_t^\perp}(s_m, a_m) \left( \pi_t(a|s) \sum_{h=m+1}^{h(s)-1} \sum_{w_h' \in W_{t,h}^\perp} \epsilon_t^\star(w_h') q^{P, \pi_t}(s_h', a_h'|s_{m+1}) \right)$$

$$\leq \sum_{m=0}^{h(s)-1} \sum_{w_m \in W_m^{\Pi,\epsilon}} \epsilon_t^\star(w_m) q^{P, \pi_t}(s_m, a_m) q^{P, \pi_t}(s, a|s_{m+1})$$

$$+ \sum_{m=0}^{h(s)-1} \sum_{w_m \in W_m^{\Pi,\epsilon}} \epsilon_t^\star(w_m) q^{P, \pi_t}(s_m, a_m) \left( \pi_t(a|s) \sum_{h=m+1}^{h(s)-1} \sum_{w_h' \in W_h^{\Pi,\epsilon}} \epsilon_t^\star(w_h') q^{P, \pi_t}(s_h', a_h'|s_{m+1}) \right)$$

The last inequality is based on Lemma 5.4, i.e., $W_{t,h}^\perp \in W_h^{\Pi,\epsilon}$ for all $t \in [T]$ and $h \in [H]$ with probability $1 - \delta$. Following the proof in Jin et al. [2019], we can take the sum over states $s \in \mathcal{S}_t^\perp \setminus \{s_h^\perp\}_{h \in [H]}$ and actions $a \in \mathcal{A}$.

$$\sum_{t=1}^{T} \sum_{s \in \mathcal{S}_t^\perp \setminus \{s_h^\perp\}_{h \in [H]}, a} \left| q^{P_t^s, \pi_t^\perp}(s, a) - q^{P_t^\perp, \pi_t^\perp}(s, a) \right|$$

$$\leq \sum_{t=1}^{T} \sum_{s \in \mathcal{S}^{\Pi,\epsilon}, a} \sum_{m=0}^{h(s)-1} \sum_{w_m \in W_m^{\Pi,\epsilon}} \epsilon_t^\star(w_m) q^{P,\pi_t}(s_m, a_m) q^{P,\pi_t}(s, a | s_{m+1})$$

$$+ \sum_{t=1}^{T} \sum_{s \in \mathcal{S}^{\Pi,\epsilon}, a} \sum_{m=0}^{h(s)-1} \sum_{w_m \in W_m^{\Pi,\epsilon}} \epsilon_t^\star(w_m) q^{P,\pi_t}(s_m, a_m) \left( \pi_t(a|s) \sum_{h=m+1}^{h(s)-1} \sum_{w_h' \in W_h^{\Pi,\epsilon}} \epsilon_t^\star(w_h') q^{P,\pi_t}(s_h', a_h' | s_{m+1}) \right)$$

$$\leq \sum_{t=1}^{T} \sum_{k<H} \sum_{m=0}^{k-1} \sum_{w_m \in W_m^{\Pi,\epsilon}} \epsilon_t^\star(w_m) q^{P,\pi_t}(s_m, a_m) \sum_{s \in \mathcal{S}_h^{\Pi,\epsilon}, a} q^{P,\pi_t}(s, a | s_{m+1})$$

$$+ \sum_{t=1}^{T} \sum_{k<H} \sum_{s \in \mathcal{S}^{\Pi,\epsilon}, a} \sum_{m=0}^{k-1} \sum_{w_m \in W_m^{\Pi,\epsilon}} \sum_{h=m+1}^{k-1} \sum_{w_h' \in W_h^{\Pi,\epsilon}} \epsilon_t^\star(w_m) q^{P,\pi_t}(s_m, a_m) \epsilon_t^\star(w_h') q^{P,\pi_t}(s_h', a_h' | s_{m+1}) \sum_{s \in \mathcal{S}_k^{\Pi,\epsilon}, a} \pi_t(a|s)$$

$$\leq \sum_{t=1}^{T} \sum_{k<H} \sum_{m=0}^{k-1} \sum_{w_m \in W_m^{\Pi,\epsilon}} \epsilon_t^\star(w_m) q^{P,\pi_t}(s_m, a_m)$$

$$+ \sum_{0 \leq m < h < k < H} |\mathcal{S}_k^{\Pi,\epsilon}| \sum_{t=1}^{T} \sum_{w_m \in W_m^{\Pi,\epsilon}} \sum_{h=m+1}^{k-1} \sum_{w_h' \in W_h^{\Pi,\epsilon}} \epsilon_t^\star(w_m) q^{P,\pi_t}(s_m, a_m) \epsilon_t^\star(w_h') q^{P,\pi_t}(s_h', a_h' | s_{m+1})$$

$$\leq \sum_{t=1}^{T} \sum_{k<H} \sum_{m=0}^{k-1} \sum_{w_m \in W_m^{\Pi,\epsilon}} \epsilon_t^\star(w_m) q^{P,\pi_t}(s_m, a_m)$$

$$+ |\mathcal{S}^{\Pi,\epsilon}| \sum_{0 \leq m < h < H} \sum_{t=1}^{T} \sum_{w_m \in W_m^{\Pi,\epsilon}} \sum_{w_h' \in W_h^{\Pi,\epsilon}} \epsilon_t^\star(w_m) q^{P,\pi_t}(s_m, a_m) \epsilon_t^\star(w_h') q^{P,\pi_t}(s_h', a_h' | s_{m+1}).$$

Here, the technical part of the proof has completed. It can be noted that the form of the right hand side of the above is exactly the same as the corresponding formula in the proof of Jin et al. [2019], Lee et al. [2020], except that we have reduced the state space from $\mathcal{S}$ to $\mathcal{S}^{\Pi,\epsilon}$. Since $\mathcal{S}^{\Pi,\epsilon}$ is $\mathcal{F}_0$-measurable, the concentration inequalities used in previous works still hold in our proof. Furthermore, compared to the confidence width defined in Jin et al. [2019], our $\epsilon_t^\star(s_{m+1}|s_m, a_m)$ has only increased by a burn-in term of order $\mathcal{O}(|\mathcal{S}^\Pi| / \max\{1, N_t(s_m, a_m)\})$. This ensures that the term in the final regret bound of order $\tilde{\mathcal{O}}(\sqrt{T})$ will not depend on $|\mathcal{S}^\Pi|$ polynomially. For the completeness of the proof, with the help of Lemma C.6 in Lee et al. [2020], we can derive a regret bound with the dependence on all parameters explicit, i.e.,

$$\sum_{t=1}^{T} \sum_{s \in \mathcal{S}_t^\perp \setminus \{s_h^\perp\}_{h \in [H]}, a} \left| q^{P_t^s, \pi_t^\perp}(s, a) - q^{P_t^\perp, \pi_t^\perp}(s, a) \right|$$

$$\leq \mathcal{O} \left( H |\mathcal{S}^{\Pi,\epsilon}| \sqrt{|\mathcal{A}| T \log \left( \frac{|\mathcal{S}^\Pi||\mathcal{A}|T}{\delta} \right)} + |\mathcal{S}^\Pi||\mathcal{S}^{\Pi,\epsilon}|^4 |\mathcal{A}| \log^2 \left( \frac{|\mathcal{S}^\Pi||\mathcal{A}|T}{\delta} \right) + |\mathcal{S}^\Pi||\mathcal{S}^{\Pi,\epsilon}|^5 |\mathcal{A}|^2 \log \left( \frac{|\mathcal{S}^\Pi||\mathcal{A}|T}{\delta} \right) \right).$$

Summing up to the first term and $(2)$, we finally upper bound $\mathbb{R}(T)$ by

$$\mathcal{O} \left( H |\mathcal{S}^{\Pi,\epsilon}| \sqrt{|\mathcal{A}| T \log \left( \frac{|\mathcal{S}^\Pi||\mathcal{A}|T}{\delta} \right)} + |\mathcal{S}^\Pi||\mathcal{S}^{\Pi,\epsilon}|^4 |\mathcal{A}| \log^2 \left( \frac{|\mathcal{S}^\Pi||\mathcal{A}|T}{\delta} \right) + |\mathcal{S}^\Pi||\mathcal{S}^{\Pi,\epsilon}|^5 |\mathcal{A}|^2 \log \left( \frac{|\mathcal{S}^\Pi||\mathcal{A}|T}{\delta} \right) + \epsilon H |\mathcal{S}^\Pi| T \right).$$

## D Proof of Auxiliary Lemmas and Corollaries

### D.1 Proof of Lemma B.1

Without loss of generality, we assume the confidence level $\delta$ is $\mathcal{F}_0$-measurable. We first note that

$$\mathbb{E}\left[\exp(X_t - 2P_t)|\mathcal{F}_{t-1}\right] \leq \mathbb{E}\left[1 + (X_t - 2P_t) + (X_t - 2P_t)^2)|\mathcal{F}_{t-1}\right] = \mathbb{E}\left[1 - P_t + X_t^2|\mathcal{F}_{t-1}\right] \leq 1,$$
$$\mathbb{E}\left[\exp(P_t - 2X_t)|\mathcal{F}_{t-1}\right] \leq \mathbb{E}\left[1 + (P_t - 2X_t) + (P_t - 2X_t)^2)|\mathcal{F}_{t-1}\right] = \mathbb{E}\left[1 - P_t + X_t^2|\mathcal{F}_{t-1}\right] \leq 1$$

where the first inequality is due to $\exp(x) \leq 1 + x + x^2$ for $x \in [-1, 1]$. Denote by $Y_n = \exp(\sum_{t=1}^{n}(X_t - 2P_t))$ and $Z_n = \exp(\sum_{t=1}^{n}(P_t - 2X_t))$, it suffices to note that both $Y_1, \ldots, Y_T$ and $Z_1, \ldots, Z_T$ are non-negative supermartingales. By Ville's inequality, we immediately have

$$\mathbb{P}\left(\exists n > 0, Y_n \geq \frac{1}{\delta}\right) \leq \delta, \ \mathbb{P}\left(\exists n > 0, Z_n \geq \frac{1}{\delta}\right) \leq \delta$$

After taking log on both $Y_n$ and $Z_n$, we get

$$\mathbb{P}\left(\exists n > 0, \sum_{t=1}^{n}(X_t - 2P_t) \geq \log\frac{1}{\delta}\right) \leq \delta, \ \mathbb{P}\left(\exists n > 0, \sum_{t=1}^{n}(P_t - 2X_t) \geq \log\frac{1}{\delta}\right) \leq \delta$$

which completes the proof.

## D.2 Proof of Lemma C.1

Given $t \in [T]$ and $h \in [H]$ and $(s, a, s') \in \mathcal{S}_h^{\perp} \setminus \{s_h^{\perp}\} \times \mathcal{A} \times \mathcal{S}_{h+1}^{\perp} \setminus \{s_{h+1}^{\perp}\}$. Notice that $t \geq t(s)$ and $t \geq t(s')$ by definition. Under the event of Lemma 6.1, it suffices to prove $|\mathcal{I}_t^1(s'|s, a) \cap \mathcal{I}_t^2(s'|s, a)| \leq \epsilon_t^{\star}(s'|s, a)$. We discuss case by case.

1. $(t(s') > t(s)$ and $N_t(s, a) \geq 2N_{t(s')}(s, a))$: In this case, there is $N_{t(s,s')}(s, a) = N_{t(s')}(s, a) \leq N_t(s, a)/2$. Thus we have

   $\bar{P}_t^{t(s,s')}(s'|s, a)$

   $\leq P_t^{\perp}(s'|s, a) + 4\sqrt{\dfrac{\bar{P}_t^{t(s,s')}(s'|s, a) \log\left(\frac{t}{\delta(s,a,s')}\right)}{\max\left\{N_t(s, a) - N_{t(s,s')}(s, a) - 1, 1\right\}}} + \dfrac{20 \log\left(\frac{t}{\delta(s,a,s')}\right)}{\max\left\{N_t(s, a) - N_{t(s,s')}(s, a) - 1, 1\right\}}$

   $\leq P_t^{\perp}(s'|s, a) + 4\sqrt{\dfrac{\bar{P}_t^{t(s,s')}(s'|s, a) \log\left(\frac{t}{\delta(s,a,s')}\right)}{\max\left\{N_t(s, a) - N_t(s, a)/2 - 1, 1\right\}}} + \dfrac{20 \log\left(\frac{t}{\delta(s,a,s')}\right)}{\max\left\{N_t(s, a) - N_t(s, a)/2 - 1, 1\right\}}$

   $\leq P(s'|s, a) + 8\sqrt{\dfrac{\bar{P}_t^{t(s')}(s'|s, a) \log\left(\frac{t}{\delta(s,a,s')}\right)}{\max\left\{N_t(s, a) - 2, 1\right\}}} + \dfrac{40 \log\left(\frac{t}{\delta(s,a,s')}\right)}{\max\left\{N_t(s, a) - 2, 1\right\}}$

   Viewing this as a quadratic inequality of $\sqrt{\bar{P}_t^{t(s')}(s'|s, a)}$, we have

   $$\sqrt{\bar{P}_t^{t(s')}(s'|s, a)} \leq \mathcal{O}\left(\sqrt{P(s'|s, a)} + \sqrt{\dfrac{\log\left(\frac{t}{\delta(s,a,s')}\right)}{\max\left\{N_t(s, a) - 2, 1\right\}}}\right).$$

   This leads to

   $$|\mathcal{I}_t^1(s'|s, a)| \leq \mathcal{O}\left(\sqrt{\dfrac{P(s'|s, a) \log\left(\frac{t}{\delta(s,a,s')}\right)}{\max\left\{N_t(s, a), 1\right\}}} + \dfrac{\log\left(\frac{t}{\delta(s,a,s')}\right)}{\max\left\{N_t(s, a), 1\right\}}\right).$$

2. $(t(s') > t(s)$ and $N_t(s, a) < 2N_{t(s')}(s, a))$: By the definition of $\mathcal{I}_t^2(s'|s, a)$, we have

   $$|\mathcal{I}_t^2(s'|s, a)| \leq \dfrac{2|\mathcal{S}^{\Pi}| + 26 \log\left(\frac{t}{\delta(s,a)}\right)}{\max\{N_{t(s')}(s, a) - 1, 1\}} \leq \mathcal{O}\left(\dfrac{|\mathcal{S}^{\Pi}| + \log\left(\frac{t}{\delta(s,a)}\right)}{\max\{N_t(s, a), 1\}}\right).$$

3. $(t(s') \leq t(s))$: When $t(s') \leq t(s)$, we have $N_{t(s,s')}(s, a) = N_{t(s)}(s, a) \leq 1$, thereby $N_{t(s,s')}(s, a)$ is also on the same order of $N_t(s, a)$. Using a similar proof as the first case, it suffices to obtain

   $$|\mathcal{I}_t^1(s'|s, a)| \leq \mathcal{O}\left(\sqrt{\dfrac{P(s'|s, a) \log\left(\frac{t}{\delta(s,a,s')}\right)}{\max\left\{N_t(s, a), 1\right\}}} + \dfrac{\log\left(\frac{t}{\delta(s,a,s')}\right)}{\max\left\{N_t(s, a), 1\right\}}\right).$$

Using the above we complete the proof.

## D.3 Details for Corollary C.2

As in Jin et al. [2019], we decompose the regret into four terms

$$\mathbb{R}^{\text{UOB-REPS}}(K) = \sum_{k=1}^{K} \langle q^{P,\pi_k} - q^{\hat{P}_k,\pi_k}, \ell_k \rangle + \sum_{k=1}^{K} \langle q^{\hat{P}_k,\pi_k}, \ell_k - \hat{\ell}_k \rangle$$

$$= \sum_{k=1}^{K} \langle q^{\hat{P}_k,\pi_k} - q^{P,\pi_\star}, \hat{\ell}_k \rangle + \sum_{k=1}^{K} \langle q^{P,\pi_\star}, \hat{\ell}_k - \ell_k \rangle.$$

Here, $\hat{P}_k \in \mathcal{P}_k$ is an approximation transition selected from the confidence set $\mathcal{P}_k$. $\hat{\ell}_k$ is the loss estimator, which is defined by

$$\hat{\ell}_k(s,a) = \frac{\ell_k(s,a)}{u_k(s,a) + \gamma} \mathbb{1}_k\{(s,a) \text{ is visited}\},$$

where $\gamma$ is an adaptive exploration rate and $u_k(s,a) = \max_{\hat{P} \in \mathcal{P}_k} q^{\hat{P},\pi_k}(s,a)$ is an upper bound of the probability of visiting state-action pair $(s,a)$ with confidence set $\mathcal{P}_k$.

Now we show how to refine the regret bound proposed in Theorem 3 [Jin et al., 2019]. For the first term, we immediately have

$$\sum_{k=1}^{K} \langle q^{P,\pi_k} - q^{\hat{P}_k,\pi_k}, \ell_k \rangle \leq \sum_{k=1}^{K} \sum_{s \in \mathcal{S}, a \in \mathcal{A}} |q^{P,\pi_k} - q^{\hat{P}_k,\pi_k}||\ell_k(s,a)|$$

$$\leq \sum_{k=1}^{K} \sum_{s \in \mathcal{S}, a \in \mathcal{A}} |q^{P,\pi_k} - q^{P_k^s,\pi_k}||\ell_k(s,a)|$$

by the definition of $P_k^s$. For the second term, there is

$$\sum_{k=1}^{K} \langle q^{\hat{P}_k,\pi_k}, \ell_k - \hat{\ell}_k \rangle = \sum_{k=1}^{K} \langle q^{\hat{P}_k,\pi_k}, \ell_k - \mathbb{E}[\hat{\ell}_k] \rangle + \sum_{k=1}^{K} \langle q^{\hat{P}_k,\pi_k}, \mathbb{E}[\hat{\ell}_k] - \hat{\ell}_k \rangle.$$

Since $\langle q^{\hat{P}_k,\pi_k}, \hat{\ell}_k \rangle \leq H$ for sure, applying Azuma's inequality we have $\sum_{k=1}^{K} \langle q^{\hat{P}_k,\pi_k}, \mathbb{E}[\hat{\ell}_k] - \hat{\ell}_k \rangle \leq H\sqrt{2K \log(1/\delta)}$. It suffices to focus on $\sum_{k=1}^{K} \langle q^{\hat{P}_k,\pi_k}, \ell_k - \mathbb{E}[\hat{\ell}_k] \rangle$.

$$\sum_{k=1}^{K} \langle q^{\hat{P}_k,\pi_k}, \ell_k - \mathbb{E}[\hat{\ell}_k] \rangle = \sum_{k=1}^{K} \sum_{s \in \mathcal{S}, a \in \mathcal{A}} q^{\hat{P}_k,\pi_k}(s,a)\ell_k(s,a) \left( 1 - \frac{q^{P,\pi_k}(s,a)}{u_k(s,a) + \gamma} \right)$$

$$\leq \sum_{k=1}^{K} \sum_{s \in \mathcal{S}, a \in \mathcal{A}} \frac{q^{\hat{P}_k,\pi_k}(s,a)}{u_k(s,a) + \gamma}\ell_k(s,a) \left( u_k(s,a) + \gamma - q^{P,\pi_k}(s,a) \right)$$

$$\leq \sum_{k=1}^{K} \sum_{s \in \mathcal{S}, a \in \mathcal{A}} |u_k(s,a) - q^{P,\pi_k}(s,a)||\ell_k(s,a)| + \gamma|\mathcal{S}||\mathcal{A}|T.$$

Notice that

$$u_k(s,a) = \max_{\hat{P} \in \mathcal{P}_k} q^{\hat{P},\pi_k}(s,a) = \pi_k(a|s) \max_{\hat{P} \in \mathcal{P}_k} q^{\hat{P},\pi_k}(s).$$

Therefore, denote by $\hat{P} = \arg\max_{\hat{P} \in \mathcal{P}_k} q^{\hat{P},\pi_k}(s)$, we have

$$\sum_{a \in \mathcal{A}} |u_k(s,a) - q^{P,\pi_k}(s,a)||\ell_k(s,a)| \leq \sum_{a \in \mathcal{A}} |q^{\hat{P},\pi_k}(s,a) - q^{P,\pi_k}(s,a)||\ell_k(s,a)|$$

$$\leq \sum_{a \in \mathcal{A}} |q^{P_k^s,\pi_k}(s,a) - q^{P,\pi_k}(s,a)||\ell_k(s,a)|$$

by the definition of $P_k^s$.

For the third and fourth terms, the proof completely follow the proof of Lemma 12 and 14 in Jin et al. [2019], i.e.,

$$\sum_{k=1}^{K} \langle q^{\hat{P}_k, \pi_k} - q^{P, \pi_\star}, \hat{\ell}_k \rangle \leq \mathcal{O}\left(\frac{H \ln(|\mathcal{S}||\mathcal{A}|)}{\eta} + \eta|\mathcal{S}||\mathcal{A}|K + \frac{\eta H \ln(H/\delta)}{\gamma}\right)$$

and

$$\sum_{k=1}^{K} \langle q^{P, \pi_\star}, \hat{\ell}_k - \ell_k \rangle \leq \mathcal{O}\left(\frac{H \ln(|\mathcal{S}||\mathcal{A}|/\delta)}{\gamma}\right).$$

By setting $\eta = \gamma = \sqrt{\frac{H \log(H|\mathcal{S}||\mathcal{A}|/\delta)}{|\mathcal{S}||\mathcal{A}|K}}$, we finally upper bound $\mathbb{R}^{\text{UOB-REPS}}(K)$ by

$$\mathcal{O}\left(\sqrt{H|\mathcal{S}||\mathcal{A}|K \log\left(\frac{|\mathcal{S}||\mathcal{A}|K}{\delta}\right)} + \sum_{k=1}^{K} \sum_{s \in \mathcal{S}, a \in \mathcal{A}} |q^{P_k^s, \pi_k}(s, a) - q^{P, \pi_k}(s, a)| |\ell_k(s, a)|\right).$$

Notice that the proof above requires tuning the learning and exploration rate in terms of $K$. To remove the restriction, a standard method is to let learning rate and exploration rate be adaptive, i.e, $\eta_k = \gamma_k = \sqrt{\frac{H \log(H|\mathcal{S}||\mathcal{A}|/\delta)}{|\mathcal{S}||\mathcal{A}|k}}$. Using the adaptive rates and taking a union bound over all $k$, we can obtain the results in Corollary C.2.

