# OpenReview forum: "State-free Reinforcement Learning"
_NeurIPS.cc/2024/Conference — NeurIPS 2024 poster_

### Official Review · Reviewer_zatZ · 2024-07-11

**Soundness:** 4
**Presentation:** 1
**Contribution:** 4
**Rating:** 7
**Confidence:** 4

**Summary:**

The paper proposes a black-box approach to make any no-regret algorithm to a state-free algorithm. The topic this paper works on is a very important topic. The theoretical results in this paper are significant, if correct. That said, the algorithm description is hard to follow, and hence, I could not verify if the results are correct or not.

**Strengths:**

Introduction and Related Work are both well-written and useful to understand what the issue in current RL theory is. The topic this paper works on is a very important topic. The theoretical results in this paper are significant, if correct.

**Weaknesses:**

That said, the algorithm proposed in this paper is not easy to understand. I tried to understand it by reading its description multiple times, but still I do not really understand it. It is because there are some undefined symbols and ambiguous definitions. For example
- What is $π^\perp$?
- What do "compatible with $\Pi$" and "it extends the pruned policy $\pi^\perp_t$ to $\pi_t$" in Line 185-187 mean?
- What is the initial set of $\mathcal{S}^\perp$?
- What are inputs and outputs of $\mathrm{ALG}$?
- Figure 1 is not really helpful to understand how the pruned space, dual trajectory, etc are obtained.

**Minor Comment**
- A paper by [Fruit et al. (2018)](https://arxiv.org/abs/1807.02373) seems related in that it also find unreachable states. I think it should be cited.

# After rebuttal review
I read the paper again and now understand the algorithm. The contribution of the paper is significant, and I highly recommend acceptance to the AC.

**Questions:**

I listed my questions in Weakness section.

**Limitations:**

The paper is mostly theoretical, and it has no potential negative societal impact.

---

> ### Author Rebuttal · Authors · 2024-08-06
>
> Thank you for the comment! We believe there are some misunderstandings, and have clarified them below. We hope the reviewer could reevaluate our paper, and are very happy to respond more questions during the reviewer-author discussion period.
>
> **Q1**:     What is $\pi^\bot$?
>
> **R1**:     $\pi^\bot$ represents a fixed default policy defined on state space $\mathcal{S}$. It can be understood as ``select an action arbitrarily" (Line 4 of Algorithm 1).
>
> **Q2**:     What does ``by playing an arbitrary action on states not in $\mathcal{S}^\bot$ compatible with $\Pi$,  it extends the pruned policy $\pi_t^\bot$ to $\pi_t$'' mean?
>
> **R2**: Note that $\pi_t^\bot$ is a function with domain $\mathcal{S}^\bot$, where an available policy $\pi_t$ needs to a function with domain $\mathcal{S}$.
> In this regard, given $\pi_t^\bot$, we need to ``lift'' the function domain from $\mathcal{S}^\bot$ to $\mathcal{S}$.
> This is achieved by playing an arbitrary action on states not in $\mathcal{S}^\bot$ compatible with $\Pi$.
>
> **Q3**: What is the initial set of $\mathcal{S}^\bot$?
>
> **R3**:         As we explained in Lines 182-184, $\mathcal{S}^\bot$ includes all the identified $\epsilon$-reachable states and $H$ additional auxiliary states.
>         Therefore, at the beginning of the game, since there is no identified $\epsilon$-reachable states, the initial set of $\mathcal{S}^\bot$ is $H$ additional auxiliary states.
>
> **Q4**:     What are inputs and outputs of ALG?
>
> **R4**:         As we explained in Lines 9-12 of Algorithm 1, the initial input of ALG is a state space, an action space, a policy set, and a confidence coefficient.
>         At each round, ALG outputs a policy that operates within the given state and action space.
>         Upon executing the policy, a trajectory is generated, which then serves as the new input for ALG.
>         Such an interaction process is generally applicable to all existing online MDP algorithms.
>
>
> **Q5**: Figure 1 is not really helpful to understand how the pruned space, dual trajectory, etc are obtained.
>
> **R5**:
> We are disappointed to hear our figure wasn't helpful, we would love to hear the reviewer's ideas for improvement during the discussion period. We really want to understand what was confusing about this picture because we are committed to improve our paper's presentation to enhance clarity for the camera ready version.
>
>
>
> **Q6**: Comparison of ``Near Optimal Exploration-Exploitation in Non-Communicating Markov Decision Processes''.
>
> **R6**:
> Thanks for the additional reference! We will add this in the camera-ready version.
>
> The problem studied in [1] is similar to the one considered in our work, i.e., how to achieve regret adaptive to $|\mathcal{S}^\Pi|$ instead of $|\mathcal{S}|$.
> However, [1] still requires knowing the possibly large set of available states $\mathcal{S}$.
> Specifically, as in its Theorem 1, the regret bound still has a term polynomial to the size of the large set of available states $|\mathcal{S}|$.
> In this regard, when $|\mathcal{S}|$ is large enough or infinite, its regret bound will become vacuous.
> Such a result aligns perfectly with our Observation 4.3: if the information of $\mathcal{S}$ is used as the input for the algorithm, using existing analysis framework, it is difficult to prevent the regret guarantee from a polynomial-level dependence on $|\mathcal{S}|$.
>
>
>
> **References**:
>
> [1]: Near Optimal Exploration-Exploitation in Non-Communicating Markov Decision Processes https://arxiv.org/abs/1807.02373

---

### Official Review · Reviewer_8pEE · 2024-07-26

**Soundness:** 2
**Presentation:** 3
**Contribution:** 2
**Rating:** 4
**Confidence:** 2

**Summary:**

This paper proposes a kind of parameter-free reinforcement learning where the algorithm does not need to have the information about states before interacting with the environment. To achieve this, the authors design a black-box reduction framework which can transform any existing RL algorithm for stochastic or adversarial MDPs into a state-free RL algorithm. The paper focuses on establishing the theoretical regret bound of such a black-box approach.

**Strengths:**

The paper proposes an interesting problem and a black-box framework for transforming any existing RL algorithm into a state-free algorithm.  The theoretical analysis seems sound and the technique may be of interest for the theoretical community.

**Weaknesses:**

While I understand that the focus of this paper is on theories, it could still be informative to include some toy experiments. For example, how would a non state-free algorithm behave if given "wrong" or insufficient knowledge/estimates of the state space, and how would the corresponding state-free algorithm (via the reduction) behaves.

**Questions:**

See above

**Limitations:**

yes

---

> ### Author Rebuttal · Authors · 2024-08-06
>
> Thank you for the comment! We would like to respectfully ask the reviewer to reassess our paper in light of the rebuttal. We believe the reviewer's concern was peripheral to our main contributions and would like to receive a fair assessment of our work.
>
> **Q1**:
> While I understand that the focus of this paper is on theories, it could still be informative to include some toy experiments. For example, how would a non state-free algorithm behave if given "wrong" or insufficient knowledge/estimates of the state space, and how would the corresponding state-free algorithm (via the reduction) behaves.
>
> **R1**:
>  This is a theoretical work, so there are no simulation experiments. Specifically, we are uncertain how existing non-state-free algorithms would perform in our state-free setting, making it challenging to design experiments.
>     For example, consider a non-state-free algorithm running on $\mathcal{S}'$. During some rounds, it executes a policy  $\pi$ and obtains a trajectory that includes a state $ s \notin \mathcal{S}' $. In this case, it is unclear how does the algorithm update its policy.

---

### Official Review · Reviewer_sD2S · 2024-08-02

**Soundness:** 3
**Presentation:** 3
**Contribution:** 3
**Rating:** 7
**Confidence:** 4

**Summary:**

The paper studies the problem of online reinforcement learning in the tabular setting when no prior knowledge about the size of the state space is available. Unlike existing algorithms, that usually require the size S as input parameter, the proposed algorithm is fully adaptive and its final performance scales with the set of reachable states.

**Strengths:**

* The algorithmic solution is quite elegant since it can be applied to any "basic" RL algorithm with regret guarantees.
* The final result achieves the desired removal of the dependency on S, which is replaced by the size of the reachable states.
* The result holds for both stochastic and adversarial settings and it can be extended to removing the dependency on the horizon H as well.

**Weaknesses:**

* I would encourage the authors to provide a clean comparison of the final bounds in the stochastic setting with the best available bounds. In particular, I'm wondering whether the restart leads to extra log terms.
* Related the previous point, I suggest the authors to make explicit the bounds for simple doubling trick strategies, so as to have a point of comparison.
* What is exactly the role of epsilon? It looks like it can be directly set to 0 and everything works the same.

Additional references of interest

* “Layered State Discovery for Incremental Autonomous Exploration” https://arxiv.org/pdf/2302.03789 This paper extends the seminal work of Lim and Auer on “Autonomous exploration” where the state space is possibly infinite (countable). In this paper, the authors managed to resolve an issue in the original paper and removed any dependency on the total number of states, where making the bound completely adaptive to the set of reachable states. Given the similarity between finite horizon and bounded distance exploration, I wonder whether there is any connection to draw between these two works. My impression is that there is quite a strong resemblance between the concept of pruned states and L-reachable states. The main technical difference is that in autonomous exploration the agent needs to explicitly restart to avoid getting “lost” in long episodes, whereas in finite horizon, the reset naturally happens each H steps.
* “Near Optimal Exploration-Exploitation in Non-Communicating Markov Decision Processes” https://arxiv.org/abs/1807.02373 This paper considers the case where the state space is somewhat “misspecified” (i.e., the set of available states is actually larger than the set of reachable states). In this case, the authors still require knowing the possibly large set of available states, so I’m referring to this paper more for completeness than for strict comparison.

**Questions:**

See weaknesses.

**Limitations:**

See above.

---

> ### Author Rebuttal · Authors · 2024-08-06
>
> Thank you for the instructive feedback! Below we address some of the questions raised by the reviewer.
>
> **Q1**: I would encourage the authors to provide a clean comparison of the final bounds in the stochastic setting with the best available bounds. In particular, I'm wondering whether the restart leads to extra log terms.
>
> **R1**:
> For the stochastic setting, our results suggest that existing algorithm UCBVI actually achieves weakly state-free learning, where its regret is only dependent on $|\mathcal{S}|$ on the log terms (Proposition 4.1). We also propose a straightforward technique, which effectively eliminates the logarithmic dependence on $|\mathcal{S}|$ under UCBVI framework (Proposition 4.2). These two propositions imply that there is no need to use our designed algorithm SF-RL for the stochastic setting.
>
> Furthermore, in inhomogeneous finite-horizon MDP, it seems that UCBVI has achieved asymptotically optimal regret. In this regard, Proposition 4.2 suggests that UCBVI (with small modifications) can also achieve state-free asymptotically optimal regret, which matches the best available bound. Besides, there is no restart in UCBVI, thus there is no extra log terms.
>
> **Q2**: Related the previous point, I suggest the authors to make explicit the bounds for simple doubling trick strategies, so as to have a point of comparison.
>
> **R2**:
> If the reviewer are referring to apply doubling trick on the state space, we will encounter some issues in the analysis. In Theorem 6.2, when a state $s$ is placed into $\mathcal{S}$, we initialize its corresponding high probability event ($\delta(s)$). In this case, we need to know exactly which state s refers to (i.e., the index of $s$). However, under doubling trick strategies, the states generated each time we double are void, which makes it impossible to establish the corresponding high probability events.
>
> **Q3**:
> What is exactly the role of epsilon? It looks like it can be directly set to $0$ and everything works the same.
>
> **R3**:
> Good question! $\epsilon$ controls the tradeoff between the regret of ALG and the error incurred by the barely reachable states we discard.
>     Specifically, when $|\mathcal{S}^\Pi|< \infty$, our results suggest that the optimal choice of $\epsilon$ should be $o(1/\sqrt{T})$ instead of $0$.
>     For example, by setting $\epsilon=1/T$, Theorem 6.2 achieves regret bound $\tilde{\mathcal{O}}(H|\mathcal{S}^{\Pi, 1/T}|\sqrt{|\mathcal{A}|T}+H|\mathcal{S}^{\Pi}|)$.
>     When $|\mathcal{S}^{\Pi, 1/T}|\ll |\mathcal{S}^{\Pi}|$, this regret will be much smaller than the regret under $\epsilon = 0$.
>
>
> **Q4**:
> Comparison of ``Layered State Discovery for Incremental Autonomous Exploration''.
>
> **R4**:
>     The objective of [1] is fundamentally different to the objective of our work (even if we transform the infinite horizon setting to our finite horizon setting).
>     In [1], the objective is to find **all** the incrementally L-controllable states.
>     If we transform this to our finite-horizon setting, the objective should be to find all the incrementally $\epsilon$-reachable states.
>     However, in our work, the objective is to minimize the regret.
>     In this regard, our algorithm does not require to find all $\epsilon$-reachable states.
>     Specifically, as described in Algorithm 1, if an $\epsilon$-reachable state has not been reached $\epsilon t$ times, it will not be included in $\mathcal{S}_t^\bot$.
>     This implies that in our work the pruned space $\mathcal{S}_t^\bot$ may **never** be the same to $\mathcal{S}^{\epsilon, \Pi}$, even if $t$ goes to infinite.
>
> **Q5**: Comparison of ``Near Optimal Exploration-Exploitation in Non-Communicating Markov Decision Processes''.
>
> **R5**: The problem studied in [2] is similar to the one considered in our work, i.e., how to achieve regret adaptive to $|\mathcal{S}^\Pi|$ instead of $|\mathcal{S}|$.
> However, [2] still requires knowing the possibly large set of available states $\mathcal{S}$.
> Specifically, as in its Theorem 1, the regret bound still has a term polynomial to the size of the large set of available states $|\mathcal{S}|$.
> In this regard, when $|\mathcal{S}|$ is large enough or infinite, its regret bound will become vacuous.
> Such a result aligns perfectly with our Observation 4.3: if the information of $\mathcal{S}$ is used as the input for the algorithm, using existing analysis framework, it is difficult to prevent the regret guarantee from a polynomial-level dependence on $|\mathcal{S}|$.
>
> Thanks for these additional references! We will add these comparisons in the camera-ready version.
>
>
> **References**:
>
> [1]: Layered State Discovery for Incremental Autonomous Exploration https://arxiv.org/pdf/2302.03789.
>
> [2]: Near Optimal Exploration-Exploitation in Non-Communicating Markov Decision Processes https://arxiv.org/abs/1807.02373

---

### Comment · Area_Chair_CKZD · 2024-08-02
**Emergency review**

**Summary.**
The paper studies the problem of online reinforcement learning in the tabular setting when no prior knowledge about the size of the state space is available. Unlike existing algorithms, that usually require the size S as input parameter, the proposed algorithm is fully adaptive and its final performance scales with the set of reachable states.

**Soundness.**
While I did not get the chance to go through the supplementary material in detail, the proof sketch in the main paper seems sound.

**Presentation.**
The paper is overall well written.

**Contribution.**
The technical scope is somehow limited to a niche problem, but the need of knowing of the state space is a long-standing limitation of existing theoretical online RL algorithms. As such, I think the contribution is significant enough.

**Strengths.**

* The algorithmic solution is quite elegant since it can be applied to any "basic" RL algorithm with regret guarantees.
* The final result achieves the desired removal of the dependency on S, which is replaced by the size of the reachable states.
* The result holds for both stochastic and adversarial settings and it can be extended to removing the dependency on the horizon H as well.

**Weaknesses.**

* I would encourage the authors to provide a clean comparison of the final bounds in the stochastic setting with the best available bounds. In particular, I'm wondering whether the restart leads to extra log terms.
* Related the previous point, I suggest the authors to make explicit the bounds for simple doubling trick strategies, so as to have a point of comparison.
* What is exactly the role of epsilon? It looks like it can be directly set to 0 and everything works the same.

Additional references of interest
* “Layered State Discovery for Incremental Autonomous Exploration” https://arxiv.org/pdf/2302.03789 This paper extends the seminal work of Lim and Auer on “Autonomous exploration” where the state space is possibly infinite (countable). In this paper, the authors managed to resolve an issue in the original paper and removed any dependency on the total number of states, where making the bound completely adaptive to the set of reachable states. Given the similarity between finite horizon and bounded distance exploration, I wonder whether there is any connection to draw between these two works. My impression is that there is quite a strong resemblance between the concept of pruned states and L-reachable states. The main technical difference is that in autonomous exploration the agent needs to explicitly restart to avoid getting “lost” in long episodes, whereas in finite horizon, the reset naturally happens each H steps.
* “Near Optimal Exploration-Exploitation in Non-Communicating Markov Decision Processes” https://arxiv.org/abs/1807.02373 This paper considers the case where the state space is somewhat “misspecified” (i.e., the set of available states is actually larger than the set of reachable states). In this case, the authors still require knowing the possibly large set of available states, so I’m referring to this paper more for completeness than for strict comparison.

---

### Decision · Program_Chairs · 2024-09-25

**Decision:**

Accept (poster)

**Comment:**

The paper resolves a very specific issue of a large part of tabular RL algorithms: the need of knowing the state space in advance as well as how that dependency translates into suboptimal regret bounds. The authors first identified a neat "trick" to remove completely the dependency and knowledge of S from the bounds of UCBVI in stochastic MDPs and replace it by the set of states that are actually reachable. While this relies on a relatively basic change, this result is already interesting in its own. The major contribution is then to have a general scheme to turn regret-minimization algorithms into state-free algorithms for both stochastic and adversarial settings at the same time.

There is general consensus about the result and novelty of the paper, while reviewer 8pEE raised concerns about the significance of the result, given the absence of empirical evaluation and comparison to non-reset-free algorithms. An additional concern, in my opinion, is that the current paper is limited to the tabular case, whereas most of the theoretical literature on exploration-exploitation has now moved to linear MDPs and non-linear approximation in general. In fact, the knowledge of the state space is not needed in non-tabular MDPs, although it is arguably replaced by the knowledge of the ranges of the linear parameters and the realizability assumption. This makes the paper somewhat borderline. Nonetheless, my recommendation is towards acceptance, given the novelty of the result and the resolution of an open question in the community.

I would still strongly encourage the authors to improve the current submission along the following axes:
* Make more explicit that the impact of the proposed algorithm is mostly in the adversarial setting, whereas the result in the stochastic setting is more "for the sake of generality", since a similar result can be already obtained with a minor change to UCBVI.
* Instantiate the general bounds to specific cases in order to make it easier for the reader to compare to existing non-state-free bounds.
* Clarify the role of epsilon as discussed in the rebuttal.
* Include the clarifications provided to reviewer zatZ during the rebuttal.